# PPM-Decay: A computational model of auditory prediction with memory decay

**Peter M. C. Harrison**[1,2]*, **Roberta Bianco**[3], **Maria Chait**[3], **Marcus T. Pearce**[2,4]

**1** Computational Auditory Perception Research Group, Max Planck Institute for Empirical Aesthetics, Frankfurt, Germany, **2** Cognitive Science Research Group, Queen Mary University of London, London, UK, **3** UCL Ear Institute, University College London, London, UK, **4** Department of Clinical Medicine, Aarhus University, Aarhus, Denmark

* peter.harrison@ae.mpg.de

**Data Availability Statement:** Our implementation of the PPM-Decay computational model is available at https://github.com/pmcharrison/ppm and https://doi.org/10.5281/zenodo.2620414. Raw data, analysis code, and generated outputs are

## Abstract

Statistical learning and probabilistic prediction are fundamental processes in auditory cognition. A prominent computational model of these processes is Prediction by Partial Matching (PPM), a variable-order Markov model that learns by internalizing *n*-grams from training sequences. However, PPM has limitations as a cognitive model: in particular, it has a perfect memory that weights all historic observations equally, which is inconsistent with memory capacity constraints and recency effects observed in human cognition. We address these limitations with PPM-Decay, a new variant of PPM that introduces a customizable memory decay kernel. In three studies—one with artificially generated sequences, one with chord sequences from Western music, and one with new behavioral data from an auditory pattern detection experiment—we show how this decay kernel improves the model's predictive performance for sequences whose underlying statistics change over time, and enables the model to capture effects of memory constraints on auditory pattern detection. The resulting model is available in our new open-source R package, *ppm* (https://github.com/pmcharrison/ppm).

## Author summary

Humans hear a rich variety of sounds throughout everyday life, ranging from the basic (e.g. an alarm clock, a whistling kettle, an ambulance siren) to the complex (e.g. speech, music, birdsong). Understanding these sounds depends on an ability to detect and remember patterns in these sounds, patterns ranging from the two-tone oscillation of the ambulance siren to the classic four-chord progression of Western popular music. A key challenge in audition research is to develop effective computer models of these pattern-detection processes. The Prediction by Partial Matching model is one such model, originally developed for data compression, that learns statistical patterns of varying complexity from sequences of discrete symbols (e.g. 'A, B, A, A, B, A, B'). In previous research this model has proved particularly effective for simulating listeners' responses to music as well as other kinds of auditory sequences. However, the model is an unrealistic simulation of human cognition in that it possesses a perfect memory, unbounded in capacity, where

archived at https://doi.org/10.5281/zenodo.3603058.

**Funding:** PH was supported by a doctoral studentship from the Engineering and Physical Sciences Research Council (EPSRC, https://epsrc.ukri.org/) and Arts and Humanities Research Council (AHRC, https://ahrc.ukri.org/) Centre for Doctoral Training in Media and Arts Technology (EP/L01632X/1). The funders did not play any role in the study design, data collection and analysis, decision to publish, or preparation of the manuscript.

**Competing interests:** The authors have declared that no competing interests exist.

historic events are recalled just as clearly as recent events. In this paper we therefore introduce a memory-decay component to the model, whereby the salience of historic auditory events decreases over time in line with the dynamics of human auditory memory. We present an experiment showing that this memory-decay model provides a natural account of experimental data from an auditory pattern detection task, explaining how human performance deteriorates as a function of the length of the patterns being detected and the speed at which they are played. Conversely, we also present two simulation studies showing that this memory-decay component can improve pattern detection in auditory environments whose statistical structure changes dynamically over time. These studies indicate the potential benefit of incorporating memory constraints into statistical models of auditory pattern detection, and highlight how these memory constraints can both impair and improve pattern detection, depending on the environment.

## Introduction

Humans are sensitive to structural regularities in sound sequences [1–10]. This structural sensitivity underpins many aspects of audition, including sensory processing [11, 12], auditory scene analysis [13, 14], language acquisition [15], and music perception [16].

The Prediction by Partial Matching (PPM) algorithm is a powerful approach for modeling this sensitivity to sequential structure. PPM is a variable-order Markov model originally developed for data compression [17] that predicts successive tokens in symbolic sequences on the basis of $n$-gram statistics learned from these sequences. An $n$-gram is a contiguous sequence of $n$ symbols, such as 'ABA' or 'ABB'; an $n$-gram model generates conditional probabilities for symbols, for example the probability that the observed sequence 'AB' will be followed by the symbol 'A', based on the frequencies of different $n$-grams in a training corpus. Different values of $n$ yield different operational characteristics: in particular, small values of $n$ are useful for generating reliable predictions when training data are limited, whereas large values of $n$ are useful for generating more accurate predictions once sufficient training data have been obtained. The power of PPM comes from combining together multiple $n$-gram models with different orders (i.e. different values of $n$), with the weighting of these different orders varying according to the amount of training data available. This combination process allows PPM to retain reliable performance on small training datasets while outperforming standard Markov chain models with larger training datasets.

The PPM algorithm has been adopted by cognitive scientists and neuroscientists as a cognitive model for how human listeners process auditory sequences. The algorithm has proved particularly useful in modeling music perception, forming the basis of the Information Dynamics Of Music (IDyOM) model of [18] which has been successfully applied to diverse musical phenomena such as melodic expectation [19], emotional experience [20], similarity perception [21], and boundary detection [22]. More recently, the PPM algorithm has been applied to non-musical auditory modeling, including the acquisition of auditory artificial grammars [5] and the detection of repeating patterns in fast tone sequences [3].

These cognitive studies typically use PPM as an *ideal-* or *rational- observer* model. Applied to a particular experimental paradigm, an ideal-observer model simulates a theoretically optimal strategy for performing the participant's task. This optimal strategy provides a benchmark against which human performance can be measured; deviations from this benchmark can then be analysed to yield further insights into human cognition. In artificial experimental paradigms, where the stimuli are generated according to a prespecified formal model, it is often

possible to derive a 'true' ideal-observer model that provably attains optimal performance. However, in naturalistic domains (e.g. music, language) the researcher does not typically have access to the true model that generated the stimuli, and so it is not possible to construct a provably optimal ideal-observer model. Moreover, in certain experimental paradigms [3] it is unlikely that the participant's cognitive processes reflect a strategy perfectly tailored to the exact experimental task; instead, they are likely to reflect general principles that tend to work well for naturalistic perception. PPM is typically applied in these latter contexts: it does not constitute the provably optimal observer for most particular tasks, but it represents a rational model of predictive processing that is assumed to approximate ideal performance for a broad variety of sequential stimuli.

However, the PPM algorithm suffers from an important limitation when applied to cognitive modeling. All observed data are stored in a single homogenous memory unit, with historic observations receiving equal salience to recent observations. This is problematic for two reasons. First, it means that the model performs suboptimally on sequences where the underlying statistical distribution changes over time. Second, it means that the model cannot capture how human memory separates into distinct stages with different capacity limitations and temporal profiles, and the way that these different stages interact to determine cognitive performance [23–25].

Some sequence modeling approaches from the cognitive literature do incorporate phenomena such as recency effects and capacity limits. [26] describe a hand-crafted model for expectations elicited by symbolic sequences that incorporates an exponential-decay component. [27] and [28] describe Bayesian inference approaches where continuous observations are assumed to be generated from Gaussian distributions whose underlying parameters change at unknown points, a situation where the optimal inference strategy involves downweighting historic observations. The latter paper's model additionally incorporates a fixed bound on memory capacity and a perceptual noise parameter, improving its cognitive plausibility. [29] describe a similar approach with categorical observations generated by a first-order Markov model, and [30] model similar data using both a hierarchical Bayesian model and a simpler exponential-decay model. [31] present participants with sequences of synthetic face images, and model resulting brain activity with an exponential-decay memory model. These studies demonstrate the importance of recency effects for sequence modeling; however, the resulting models generally cannot learn the kinds of higher-order statistics that PPM specializes in.

Some approaches from the natural language processing literature also incorporate recency effects. Here the motivation is primarily 'model adaptation', helping the model to respond to changing statistics in the data being modeled; a useful byproduct can be reducing the computational resources required by the implementation. A recency effect with a particularly efficient implementation is exponential decay, which has been applied to a variety of model types, such as trigram word models [32, 33], Web search term recurrence models [34], topic models [35], data streaming models [36], and hidden Markov models [37]. Related heuristics are the sliding window of the Lempel-Ziv data compressor [38] and the nonstationary update rule of the PAQ compressor [39]. However, these models have yet to be integrated into mainstream cognitive modeling research, having been primarily optimized for engineering applications rather than for cognitive plausibility.

In the specific context of PPM, some attempts have been made to implement memory bounds and recency effects. Moffat's [40] implementation allocated a fixed amount of storage space to the trie data structure used to store observed data, and rebuilt this tree from scratch each time this storage limit was exceeded, after [41]. This solution may be computationally efficient but it has limited cognitive validity. [42] introduced a technique whereby two PPM models would be trained, a long-term model and a short-term model, with the long-term model

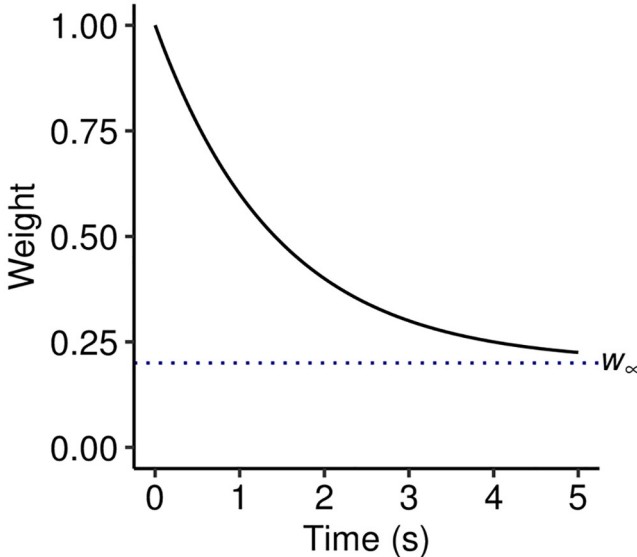

**Fig 1. A simple decay kernel.** The kernel is defined by an initial weight of $w_0 = 1$, an exponential decay with half life $t_{0.5} = 1$ s, and an asymptotic weight $w_\infty = 0.2$.

retaining training data from all historic sequences and the short-term model only retaining training data from the current sequence. The predictions from these two models would then be combined to form one probability distribution. This technique works well for capturing the distinction between the structural regularities characterizing a domain (e.g. a musical style, a language) and the statistical regularities local to a given item from the domain (e.g. a musical composition or a specific text), but it cannot capture recency effects within a given sequence or distinguish between historic sequences of different vintages.

Here we present a new version of the PPM algorithm that directly addresses these issues of memory modeling. This new algorithm, termed 'PPM-Decay', introduces a decay kernel that determines the weighting of historic data as a function of various parameters, typically the time elapsed since the historic observation, or the number of subsequent observations (Fig 1). It also introduces stochastic noise into memory retrieval, allowing the model to capture analogous imperfections in human memory. We have developed an open-source implementation of the model in C++, made available in the R package *ppm* and released under the MIT license, that allows the user to configure and evaluate different variants of the PPM-Decay model on arbitrary sequences.

We demonstrate the utility of this new algorithm in a series of experiments corresponding to a variety of task domains. Experiment 1 simulates the prediction of sequences generated from a prespecified statistical model, and shows that incorporating memory decay improves the predictive performance of PPM for sequences when the underlying model parameters change over time. Experiment 2 simulates the prediction of chord sequences from three musical styles, and shows that a decay profile with a non-zero asymptote is useful for capturing a combination of statistical regularities specific to the current composition and statistical regularities general to the musical style. Experiment 3 models new empirical data from human listeners instructed to detect repeated patterns in fast tone sequences, and shows that a multi-stage decay kernel is useful for explaining human performance. Together these experiments speak to the utility of the PPM-Decay algorithm as a cognitive model of symbolic sequence processing.

## Overview of the PPM-Decay model

This section provides a short overview of the PPM-Decay model. A full exposition is provided at the end of the paper.

The PPM-Decay model is a direct extension of the PPM model [17] incorporating the interpolated smoothing technique of [43] alongside a custom memory retrieval function. PPM is an incremental sequence prediction model: it progresses through sequences symbol by symbol, generating a predictive distribution for the next symbol before it arrives, with this predictive distribution derived from statistics learned from the model's previous observations. The predictive distribution comprises a list of probabilities for all symbols in the alphabet, with likely continuations receiving high probabilities and unlikely continuations receiving low probabilities.

PPM's predictions are produced by combining predictions from several sub-models, specifically *n*-gram models. An *n*-gram is a sequence of *n* adjacent symbols; for example, 'ABCABC' is an example of a 6-gram. An *n*-gram model generates predictions by counting and comparing observations of *n*-grams. To predict what happens after the sequence 'ABCAB', a 6-gram model would look at all 6-gram occurrences of the form 'ABCABX', where 'X' is allowed to vary. The predicted probability of observing 'C' having just observed 'ABCAB' would then be equal to the number of observations for 'ABCABC' divided by the total number of observations for *n*-grams of the form 'ABCABX'.

The *Markov order* of an *n*-gram model is equal to the length of its *n*-grams minus one. Different orders of *n*-gram model have different strengths and weaknesses. Low-order *n*-gram models are quick to learn, but can only capture simple structure; high-order *n*-gram models can capture more complex structure, but require more training data. The power of PPM comes from flexibly switching between different Markov orders based on context; this is why it is termed a 'variable-order' Markov model.

Different PPM variants use different techniques for combining *n*-gram models. The original PPM model used a technique called backoff smoothing, where the model chooses one top-level *n*-gram model from which to generate predictions for a given context, and only calls upon lower-level *n*-gram models in the case of symbols that have never been seen before by the top-level *n*-gram model. Here we instead use the interpolated smoothing technique of [43], where lower-order models contribute to predictions for *all* the symbols in the alphabet. This approach is more computationally expensive, but results in significantly improved predictive power.

The different *n*-gram models are weighted according to training data availability for the current predictive context: with limited training data, low-order models receive the most weight, but as training data accumulates, high-order models receive increasing weight. The procedure for deriving these weights is termed the 'escape method'; though various improved escape methods have been introduced over the years, here we retain the escape method from the original PPM model [17], because it generalizes most naturally to the memory-decay procedures considered here.

The PPM-Decay model adds a custom memory retrieval function to PPM. This retrieval function modifies how *n*-grams are counted: whereas in the original PPM model, each *n*-gram observation contributes a fixed amount to the respective *n*-gram count, in the PPM-Decay model the contribution of a given *n*-gram observation decreases as a function of the time elapsed since the observation. This function is termed the decay kernel; an example decay kernel is plotted in Fig 1. In the following experiments we explore how decay kernels may be constructed that help predictive performance for particular kinds of sequences, and that reflect the dynamics of human auditory memory.

## Results

### Experiment 1: Memory decay helps predict sequences with changing statistical structure

The original PPM algorithm weights all historic observations equally when predicting the next symbol in a sequence. This represents an implicit assertion that all historic observations are equally representative of the sequence's underlying statistical model. However, if the sequence's underlying statistical model changes over time, then older observations will be less representative of the current statistical model than more recent observations. In such scenarios, an ideal observer should allocate more weight to recent observations than historic observations when predicting the next symbol.

Various weighting strategies can be envisaged representing different inductive biases about the evolution of the sequence's underlying statistical model. A useful starting point is an exponential weighting strategy, whereby an observation's salience decreases by a constant fraction every time step. Such a strategy is biologically plausible in that the system does not need to store a separate trace for each historic observation, but instead can simply maintain one trace for each statistical regularity being monitored (e.g. one trace per $n$-gram), which is incremented each time the statistical regularity is observed and decremented automatically over time. This exponential-weighting strategy can also be rationalised as an approximation to optimal Bayesian weighting for certain types of sequence structures [44].

We will now describe a proof-of-concept experiment to demonstrate the intuitive notion that such weighting strategies can improve predictive performance in the PPM algorithm. This experiment used artificial symbolic sequences generated from an alphabet of five symbols, where the underlying statistical model at any particular point in time was defined by a first-order Markov chain. A first-order Markov chain defines the probability of observing each possible symbol conditioned on the immediately preceding symbol; second-order Markov chains are Markov chains that take into account two preceding symbols, whereas zeroth-order Markov chains take into account zero preceding symbols. Our sequence-generation models were designed as hybrids between zeroth-order and first-order Markov chains, reflecting PPM's capacity to model sequential structure at different Markov orders. These generative models took the form of first-order Markov chains, constructed as follows. First, we sampled five first-order conditional distributions from a symmetric Dirichlet prior with concentration parameter 0.1, with each distribution corresponding to a different conditioning symbol from the alphabet. Second, we averaged these conditional distributions with a 0th-order distribution sampled from the same Dirichlet prior. The resulting Markov chain models can be represented as two-dimensional transition matrices, where the cell in the $i$th row and the $j$th column identifies the probability of observing symbol $j$ given that the previous symbol was $i$ (Fig 2A). Zeroth-order structure is then manifested as correlations between transition probabilities in the same column, and can be summarised in marginal bar plots (Fig 2A).

Each sequence began according to an underlying statistical model constructed by the above procedure, with the first symbol in each sequence being sampled from the model's stationary distribution. At the next symbol, the underlying statistical model was either preserved with probability .99 or discarded and regenerated with probability .01. The new symbol was then sampled from the resulting statistical model conditioned on the immediately preceding symbol. This procedure was repeated to generate a sequence totalling 500 symbols in length.

Individual experimental trials were then conducted as follows. The PPM-Decay model was presented with one symbol at a time from a sequence constructed according to the procedure defined above, and instructed to return a predictive probability distribution for the next symbol. A single prediction was then extracted from this probability distribution, corresponding

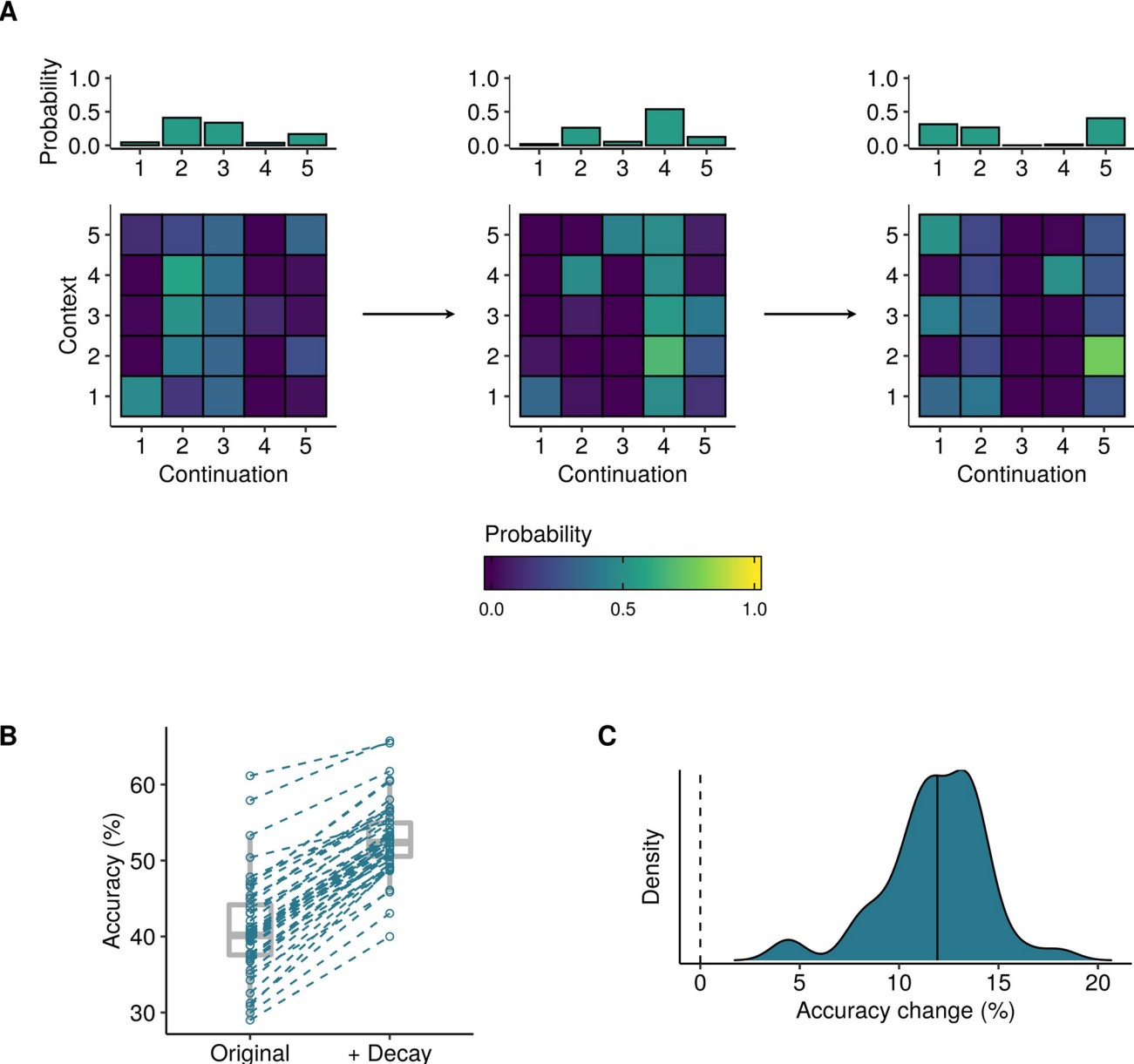

**Fig 2. Illustrative plots for Experiment 1. A)** Example sequence-generation models as randomly generated in Experiment 1. The bar plots describe 0th-order symbol distributions, whereas the matrices describe 1st-order transition probabilities. **B)** Repeated-measures plot indicating how predictive accuracy for individual sequences ($N = 500$, hollow circles) increases after the introduction of an exponential-decay kernel. **C)** Absolute changes in predictive accuracy for individual sequences, as summarised by a kernel density estimator. The median accuracy change is marked with a solid vertical line.

to the symbol assigned the highest probability. Prediction success was then operationalized as the proportion of observed symbols that were predicted correctly.

This experimental paradigm was used to evaluate a PPM-Decay model constructed with an exponential-decay kernel and a Markov order bound of one. This kernel is parametrized by a single half-life parameter, defined as the time interval for an observation's weight to decrease by 50%. This half-life parameter was optimized by evaluating the model on 500 experimental trials generated by the procedure described above, maximizing mean prediction success over all trials using Rowan's [45] Subplex algorithm as implemented in the NLopt package [46], and

refreshing the model's memory between each trial. The resulting half-life parameter was 12.26. The PPM-Decay model was then evaluated with this parameter on a new dataset of 500 experimental trials and compared with an analogous PPM model without the decay kernel.

The results are plotted in Fig 2B and 2C. They indicate that the exponential-decay kernel reliably improves the model's performance, with the median percentage accuracy increasing from 48.8% to 62.2%. The exponential-decay kernel causes the algorithm to downweight historic observations, which are less likely to be representative of the current sequence statistics, thereby helping the algorithm to develop an effective model of the current sequence statistics and hence generate accurate predictions. Correspondingly, we can say that the exponential-decay model better resembles an ideal observer than the original PPM model.

### Experiment 2: Memory decay helps predict musical sequences

We now consider a more complex task domain: chord sequences in Western music. In particular, we imagine a listener who begins with zero knowledge of a musical style, but incrementally acquires such knowledge through the course of musical exposure, and uses this knowledge to predict successive chords in chord sequences. This process of musical prediction is thought to be integral to the aesthetic experience of music, and so it is of great interest to music theorists and psychologists to understand how these predictions are generated [16, 47–50].

Chord sequences in Western music resemble sentences in natural language in the sense that they can be modeled as sets of symbols drawn from a finite dictionary and arranged in serial order. Such chord sequences provide the structural foundation of most Western music. For the purpose of modeling with the PPM algorithm, it is useful to translate these chord sequences into sequences of integers, which we do here using the mapping scheme described in Methods. For example, the first eight chords of the Bob Seger song 'Think of Me' might be represented as the integer sequence '213, 159, 33, 159, 213, 159, 33, 159'.

Here we consider chord sequences drawn from three musical corpora: a corpus of popular music sampled from the Billboard 'Hot 100' charts between 1958 and 1991 [51], a corpus of jazz standards sampled from an online forum for jazz musicians [52], and a corpus of 370 chorale harmonizations by J. S. Bach [53], translated into chord sequences using the chord labeling algorithm of [54] (see Methods for details). These three corpora may be taken as rough approximations of three musical styles: popular music, jazz music, and Bach chorale harmonizations. While we expect these three corpora each to be broadly consistent with general principles of Western tonal harmony [55], we also expect each corpus to possess distinctive statistical regularities that differentiate the harmonic languages of the three musical styles [52, 56–58]. Fig 3 displays example chord sequences from these three corpora, alongside their corresponding integer encodings.

We expect the underlying sequence statistics to vary as we progress through a musical corpus. Sequence statistics are likely to change significantly at the boundaries between compositions, but they are also likely to change within compositions, as the chord sequences modulate to different musical keys, and travel through different musical sections. Similar to Experiment 1, we might therefore hypothesize that some kind of decay kernel should help the listener maintain an up-to-date model of the sequence statistics, and thereby improve predictive performance.

However, unlike Experiment 1, the chord sequences within a given musical corpus are likely to share certain statistical regularities. If the corpus is representative of a given musical style, then these statistical regularities will correspond to a notion of 'harmonic syntax', the underlying grammar that defines the harmonic conventions of that musical style. An ideal model will presumably take advantage of these stylistic conventions. However, the exponential-decay kernel from Experiment 1 is not well-suited to this task, because observations from historic sequences continuously decay in weight until they make essentially no contribution to

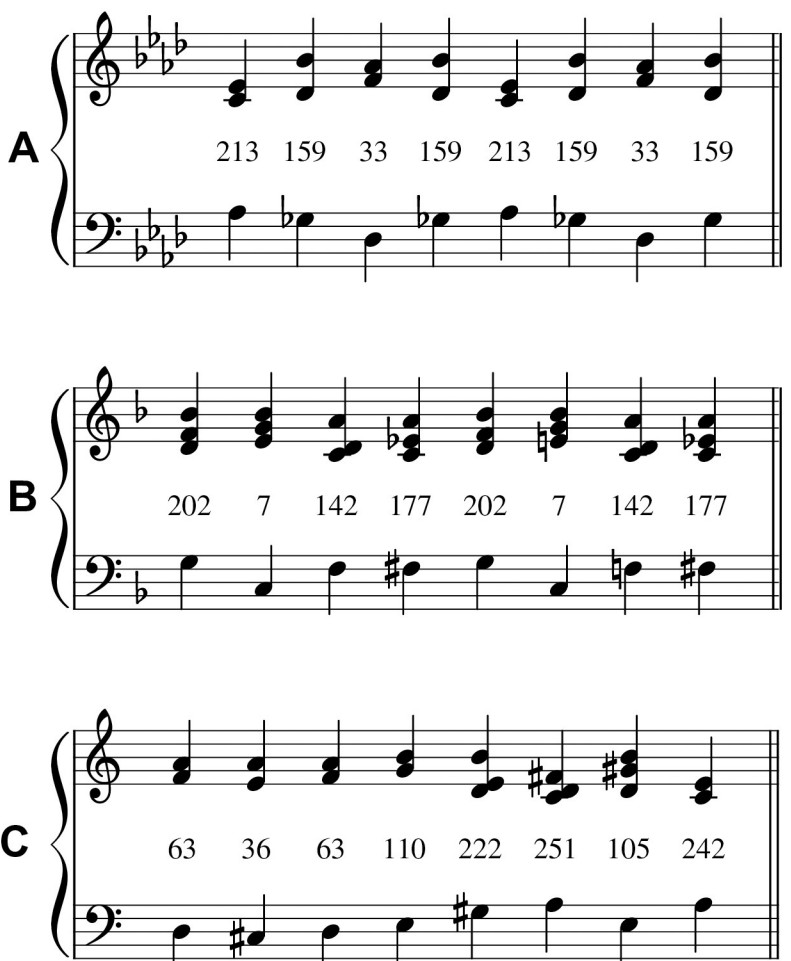

**Fig 3. Sample chord sequences analyzed in Experiment 2. A)** represents the popular music corpus ('Night Moves', by Bob Seger), **B)** represents the jazz corpus ('Thanks for the Memory', by Leo Robin), and **C)** represents the Bach chorale harmonization corpus ('Mit Fried und Freud ich fahr dahin', by J. S. Bach). Each chord is labeled by its integer encoding within the chord alphabet for the respective corpus. Each chord sequence corresponds to the first eight chords of the first composition in the downsampled corpus. Each chord is defined by a combination of a bass pitch class (lower stave) and a collection of non-bass pitch classes (upper stave). For visualization purposes, bass pitch classes are assigned to the octave below middle C, and non-bass pitch classes to the octave above middle C.

the model. This is not ideal because these historic sequences will still contribute useful information about the musical style. Here we therefore evaluate a modified exponential-decay kernel, where memory traces decay not to zero but to a positive asymptote (see e.g. Fig 1). Such a kernel should provide a useful compromise between following the statistics of the current musical passage and capturing long-term knowledge of a style's harmonic syntax.

We conducted our experiment as follows. For each musical corpus, we simulated a listener attempting to develop familiarity with the musical style by listening to one chord sequence every day, corresponding to one composition randomly selected from the corpus without repetition, for 100 days. We supposed that the listener began each chord sequence at the same time of day, so that the beginning of each successive chord sequence would be separated by 24-hour intervals, and we supposed that each chord in each chord sequence lasted one second in duration. Similar to Experiment 1, we supposed that the listener constantly tried to predict the next chord in the chord sequence, but this time we operationalized predictive success

using the cross-entropy error metric, defined as the mean negative log probability of each chord symbol as predicted by the model. This metric is more appropriate than mean success rate for domains with large alphabet sizes, such as harmony, because it assigns partial credit when the model predicts the continuation with high but non-maximal probability. A similar metric, perplexity, is commonly used in natural language processing (cross-entropy is equal to the logarithm of perplexity). We used the cross-entropy metric to evaluate two decay kernels: the exponential-decay kernel evaluated in Experiment 1, termed the 'Decay only' kernel, and a new exponential-decay kernel incorporating a positive asymptote, termed the 'Decay + long-term learning' model. We found optimal parametrizations for these kernels using the same optimizer as Experiment 1 [45], and compared the predictive performance of the resulting optimized models to a standard PPM model without a decay kernel. Each model was implemented with a Markov order bound of four, which seems to be a reasonable upper limit for the kinds of Markovian regularities present in Western tonal harmony [49, 59, 60].

Fig 4 describes the performance of these two decay kernels. Examining the results for the three different datasets, we see that the utility of different decay parameters depends on the musical style. For the popular music corpus, incorporating exponential decay improves the model's performance by c. 1.9 bits, indicating that individual compositions carry salient short-term regularities that the model can better leverage by downweighting historic observations. Introducing a non-zero asymptote to the decay kernel does not improve predictive performance on this dataset, indicating that long-term syntactic regularities contribute very little to predictive performance over and above these short-term regularities in popular music. A different pattern is observed for the jazz and Bach chorale corpora, however. In both cases, the decay-only model performs no better than the original PPM model, presumably because any improvement in capturing local statistics is penalized by a corresponding deterioration in long-term syntactic learning. However, incorporating a non-zero asymptote in the decay kernel allows the model both to upweight local statistics and still achieve long-term syntactic learning, thereby improving predictive performance by c. 1.5 bits.

These analyses have two main implications. First, they show that more advanced decay kernels are useful for producing a predictive model that better approximates ideal performance in the cognitive task of harmony prediction. The nature of these improved kernels can be related directly to the statistical structure of Western music, where compositions within a given musical style tend to be characterized by local statistical regularities, yet also share common statistical structure with other pieces in the musical style. An ideal-observer model of harmony prediction ought therefore to recognize these different kinds of statistical structure. Second, these analyses offer quantitative high-level insights into the statistical characteristics of the three musical styles. In particular, the popular music analyses found that long-term learning offered no improvement over a simple exponential-decay kernel, implying that the harmonic structure of popular music is dominated by local repetition. In contrast, both the jazz analyses and the Bach chorale analyses found that both exponential decay and long-term learning were necessary to improve from baseline performance, implying that chord progressions in these styles reflect both short-term statistics and long-term syntax to significant degrees.

## Experiment 3: Memory decay helps to explain the dynamics of auditory pattern detection

The PPM model has recently been used to simulate how humans detect recurring patterns in fast auditory sequences [3]. Barascud et al. used an experimental paradigm where participants were played fast tone sequences derived from a finite pool of tones, with the sequences organised into two sections: a random section (labelled 'RAND') and a regular section (labelled

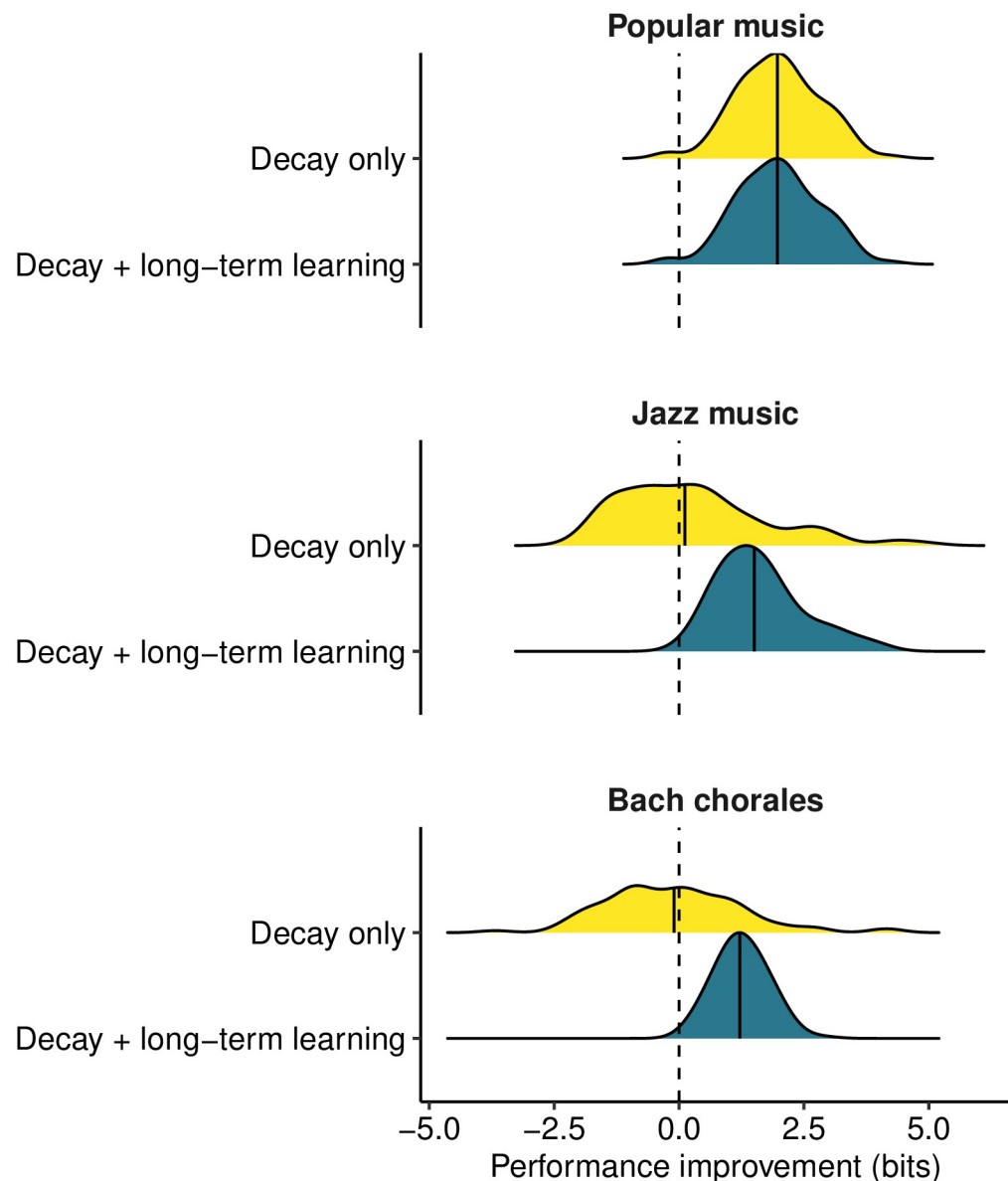

**Fig 4. Predictive performances for different decay kernels in Experiment 2.** Each composition contributed one cross-entropy value for each decay kernel; these cross-entropy values are expressed relative to the cross-entropy values of the original PPM model, and then summarised using kernel density estimators. Median performance improvements are marked with solid vertical lines.

'REG'). The random section was constructed by randomly sampling tones from the frequency pool, whereas the regular section constituted several repetitions of a fixed pattern of frequencies from the pool. These two-stage sequences, termed 'RAND-REG' sequences, were contrasted with 'RAND' sequences which solely comprised one random section. The participant's task was to detect transitions from random to regular sections as quickly as possible (see Fig 5 for an example trial).

Two simplified sequence types were also included for the purpose of baselining reaction times: 'CONT' sequences, which were simplified versions of RAND sequences that comprised just one repeated tone of a given frequency, and 'STEP' sequences, which were simplified

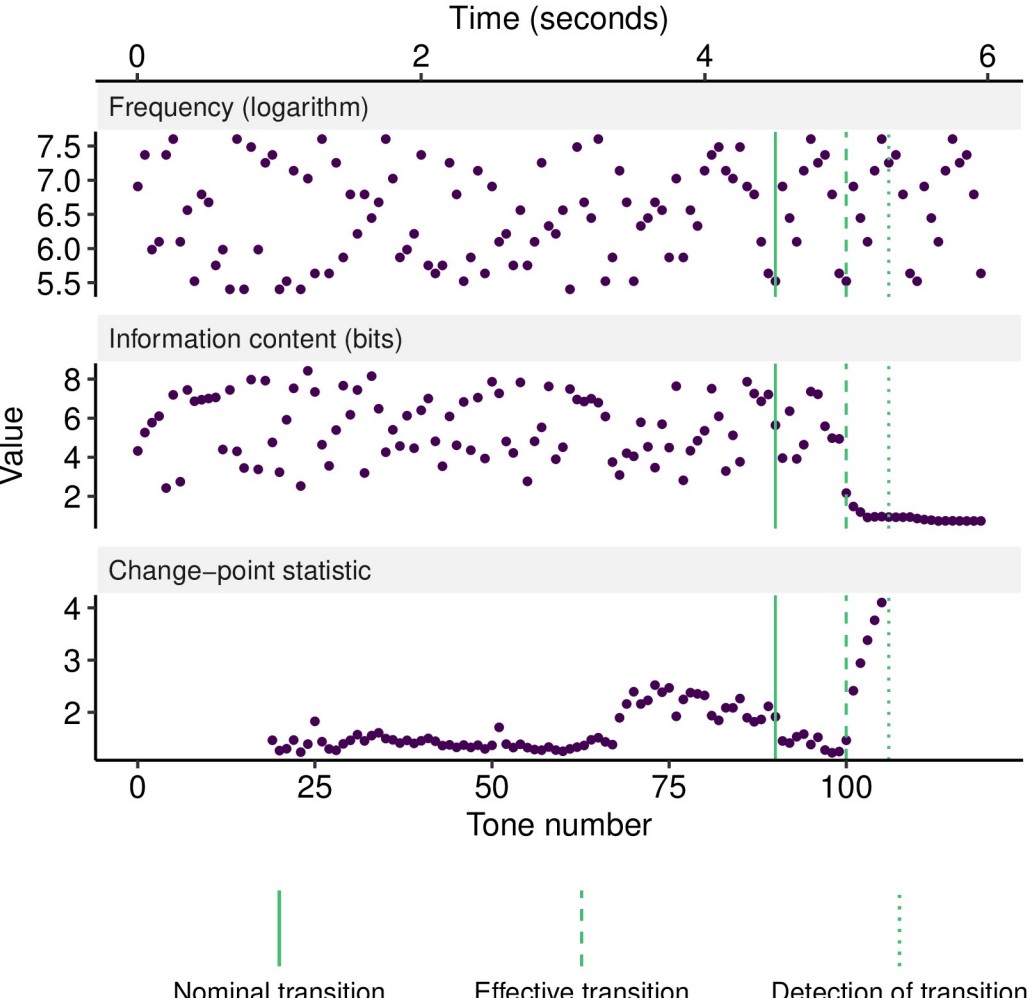

**Fig 5. Example analysis of a single trial in Experiment 3.** The three panels plot each tone's frequency, change-point statistic, and information content respectively. 'Nominal transition' denotes the point at which the pattern changes from random tones to a repeating pattern of length 10. This repetition starts to become discernible after 10 tones ('Effective transition'), at which point the sequence becomes fully deterministic. Correspondingly, information content (or 'surprise') drops, and triggers change-point detection at 'Detection of transition'.

versions of RAND-REG sequences where all tones within a given section had the same frequency, with this frequency differing between sections. We used the STEP trials to estimate baseline response times, and normalized the RAND-REG response times by subtracting these baseline response times (see Methods for more details).

These experimental stimuli were constructed according to a well-defined statistical process, and it would be straightforward to derive a model that achieves provably optimal performance on the task given a well-defined performance metric. However, Barascud et al. reasoned that the cognitive mechanisms underlying fast auditory pattern recognition would be unlikely to be tailored to exact repetition, because exact repetitions are uncommon in naturalistic auditory environments. Instead, they supposed that human performance would be better characterized by more generic regularity detection mechanisms, such as those embodied in the PPM algorithm.

In particular, [3] suggested that listeners maintain an internal predictive model of incoming tone sequences that is incrementally updated throughout each sequence, and that listeners monitor the moment-to-moment surprise experienced by this model. They modeled this process using PPM as the predictive model, and operationalized surprise as the information content of each tone, defined as the tone's negative log probability conditioned on the portion of the sequence heard so far. The authors proposed that listeners detect section changes based on the evolution of information content throughout the stimulus; in particular, changes from random to regular precipitate a sharp drop in information content, reflecting the transition from unpredictability to predictability.

Examining information content profiles produced by the PPM model, [3] concluded that an ideal observer should detect the transition from random to regular sections by the fourth tone of the second occurrence of the regular tone cycle. Analyzing behavioral and neuroimaging data, the authors found that participants reached this benchmark when the cycle length was small (5, 10 tones) but not when it was large (15, 20 tones). In other words, the ideal-observer model replicated human performance well for short cycle lengths, but some kind of cognitive constraints seemed to impair human performance for large cycle lengths.

One candidate explanation for this impaired performance is the limited capacity of auditory short-term memory. In order to detect a cycle repetition, the listener must compare incoming tones to tones that occurred at least one cycle ago. To achieve this, the listener's auditory short-term memory must therefore span at least one cycle length. Short cycles may fit comfortably in the listener's short-term memory, thereby supporting near-optimal task performance, but longer cycles may progressively test the limits of the listener's memory capacity, resulting in progressively worsened performance.

An important question is whether this memory capacity is determined by temporal limits or informational limits. A temporal memory limit would correspond to a fixed time duration, within which events are recalled with high precision, and outside of which recall performance suffers. Analogously, an informational limit would correspond to a fixed number of tones that can be recalled with high fidelity from short-term memory, with attempts to store larger numbers of tones resulting in performance detriment.

Both kinds of capacity limits have been identified for various stages of auditory memory. Auditory sensory memory, or echoic memory, is typically characterized by its limited temporal capacity but high informational capacity. Auditory working memory has a more limited informational capacity, and a temporal capacity that can be extended for long periods through active rehearsal. Auditory long-term memory seems to be effectively unlimited in both temporal and informational capacity [23–25, 61].

The auditory sequences studied by Barascud et al. used very short cycle lengths, of the order of 1 s, and were therefore likely to fall within the bounds of echoic memory. Temporal limits to echoic memory are well-documented in the literature: echoic memory seems to persist for a few seconds, and is susceptible to interference from subsequent sounds [62]. Informational limits to echoic memory are less well understood. The most relevant precedent to this paradigm in the literature seems to be [63], studying change detection in sequences of pure tones where total sequence duration ranged from 60 to 2000 ms. In this regime, change-detection performance was largely unaffected by sequence duration, and was instead constrained by the number of tones in the sequence, pointing to an informational limit to echoic memory. However, related work has studied change detection where the tones were presented simultaneously rather than sequentially, and failed to find compelling evidence for memory capacity limits over and above sensory processing limits [64, 65]. Another relevant precedent is [66], who played participants pairs of adjacent click trains each of 1 s duration, and asked participants to answer whether the second click train was a repetition of the first click train.

Performance decreased as click rate increased, which could reflect capacity limits in echoic memory, but could also reflect capacity limits in sensory processing. In summary, informational limits to echoic memory have received limited characterization in the literature, but nonetheless seem equally plausible to temporal limits as explanations of Barascud et al.'s results [3].

We conducted a behavioral experiment to test these competing explanations. We based this experiment on the regularity detection task from Barascud et al., and created six experimental conditions that orthogonalised two stimulus features: the number of tones in the cycle (10 tones or 20 tones), and the temporal duration of each tone in the cycle (25 ms, 50 ms, or 75 ms). We reasoned that if performance were constrained by informational capacity, then it would be best predicted by the number of tones in the cycle, whereas if performance were constrained by temporal limits, it would be best predicted by the total duration of each cycle. We were particularly interested in the pair of conditions with equal cycle duration but different numbers of tones per cycle (10 × 50 ms = 20 × 25 ms); a decrease in performance in the latter condition would be evidence for informational constraints on regularity detection.

The behavioral results are summarized in Fig 6. Response accuracies are plotted in Fig 6A in terms of the sensitivity metric from signal detection theory. Response accuracy was close to ceiling performance across all conditions, with the exception of the condition with the maximum-duration cycles (20 tones each of length 75 ms), where some participants fell away from ceiling performance. This is consistent with [3]; the task manipulations change how long it takes for the participant to detect the change, but they mostly do not prevent the participant from eventually detecting the change. Following [3], we therefore focus on interpreting reaction-time metrics (Fig 6B) rather than response accuracy. Here we see a clear effect of the number of tones in the cycle, with 10-tone cycles eliciting considerably lower reaction times than 20-tone cycles. This is consistent with the notion of an informational capacity to echoic memory. In particular, comparing the two conditions with equal cycle duration but different numbers of tones per cycle (10 × 50 ms tones; 20 × 25 ms tones), we see that increasing the number of tones substantially impaired performance even when cycle duration stayed constant.

Fig 6B does not show a clear effect of tone duration. However, the figure does not account for the repeated-measures structure of the data, meaning that between-condition effects may be partly masked by individual differences between participants. To achieve a more sensitive analysis, Fig 6C takes advantage of the repeated-measures structure of the data, and plots each participant's response time in the 50-ms and 75-ms conditions relative to their response time in the relevant 25-ms condition. Here we again see null or limited effects of tone duration, except in the case of the maximum-duration condition (20 tones each of length 75 ms), where reaction times seem higher than in the corresponding 25-ms and 50-ms conditions. We tested the reliability of this effect by computing each participant's difference in mean response time between the 25-ms and 75-ms conditions for the 20-tone cycles, and subtracting the analogous difference in response times for the 10-tone cycles, in other words: {RT(75 ms, 20 tones) − RT(25 ms, 20 tones)} − {RT(75 ms, 10 tones) − RT (25 ms, 10 tones)}. This number summarizes the extent to which increasing tone duration has a stronger effect on reaction times for cycles containing more tones. Using the bias-corrected and accelerated bootstrap [67], the 95% confidence interval for this parameter was found to be [2.08, 5.93]. The lack of overlap with zero indicates that the effect was fairly reliable: increasing tone duration from 25-ms and 75-ms had a stronger negative effect on reaction times for 20-tone cycles than for 10-tone cycles. This interaction between tone duration and cycle length was also evident from a 3 x 2 repeated-measures ANOVA ($F(2, 44) = 8.76$, $\eta^2 = .07$, $p < .001$).

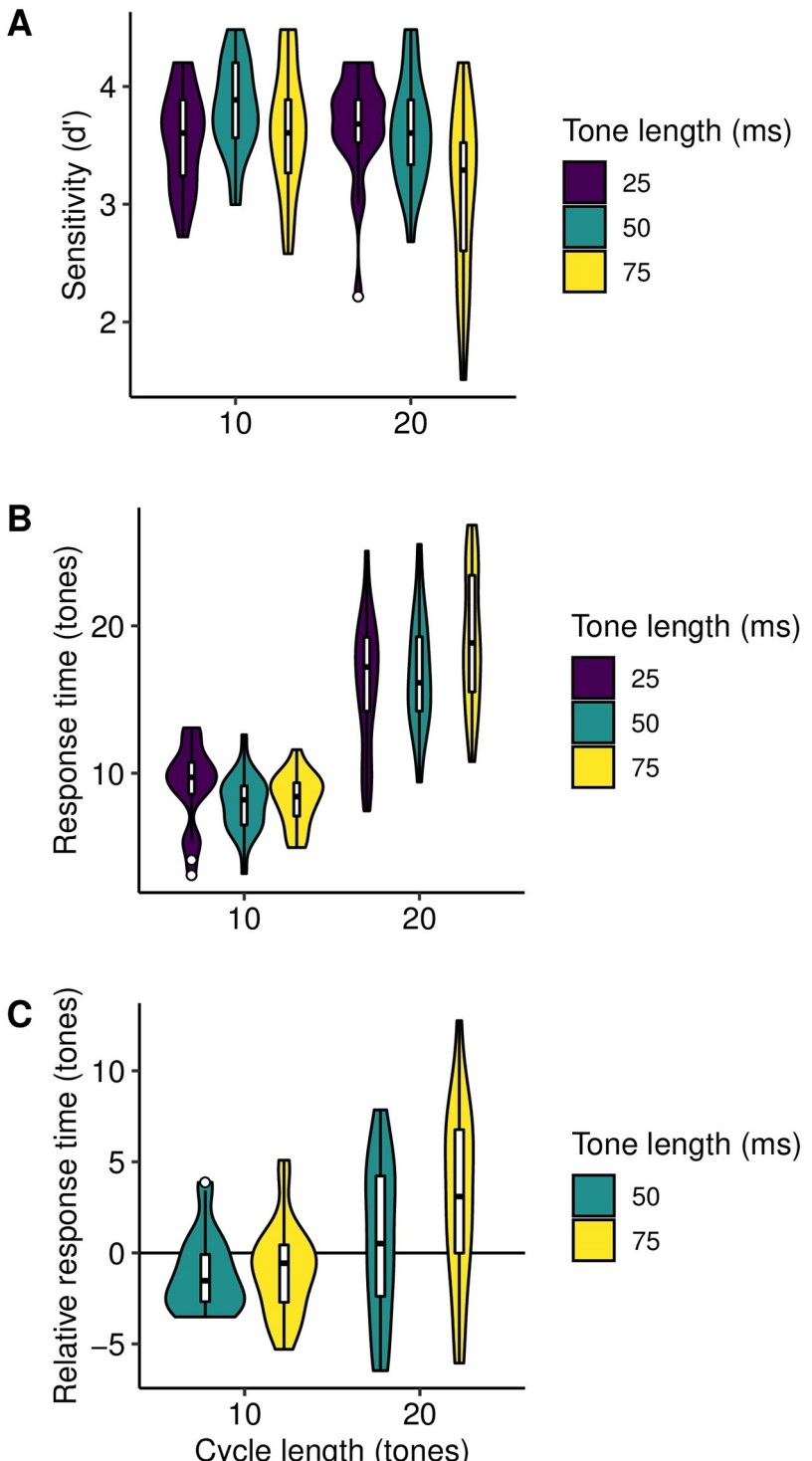

**Fig 6. Behavioral results for Experiment 3. A)** Participant *d*-prime scores by condition, as summarized by violin plots and Tukey box plots. **B)** Participant mean response times by condition, as summarized by violin plots and Tukey box plots. **C)** As **B**, except benchmarking response times against the 25 ms conditions.

To summarize, then: the behavioral data indicate that performance in this regularity-detection task was primarily constrained by the number of tones in the repeating cycles, rather than their duration. However, the data do suggest a subtle negative effect of tone duration which may manifest for cycles containing large numbers of tones.

We now consider how these effects may be reproduced by incorporating memory effects into the PPM model. Instead of the decay kernel solely operating as a function of time, as in Experiments 1 and 2, it must now account for the number of tones that have been observed by the listener. Various such decay kernels are possible. Here we decided to base our decay kernel on the following psychological ideas, inspired by previous research into echoic memory [23, 24, 68]:

1. Echoic memory operates as a continuously updating buffer that stores recent auditory information.

2. While a memory remains in the buffer, it is represented with high fidelity, and is therefore a reliable source of information for regularity detection mechanisms.

3. The buffer has a limited temporal and informational capacity. Memories will remain in the buffer either until a certain time period has elapsed, or until a certain number of subsequent events has been observed.

4. Once a memory leaves the buffer, it is represented in a secondary memory store.

5. Observations in this secondary memory store contribute less strongly to auditory pattern detection, and gradually decay in salience over time, as in Experiments 1 and 2.

These principles, formalized computationally and applied to the continuous tone sequences from the behavioral experiment, result in the decay kernels described in Fig 7. In each case the buffer is limited to a capacity of 15 tones, which corresponds to a time duration of 0.375 s for

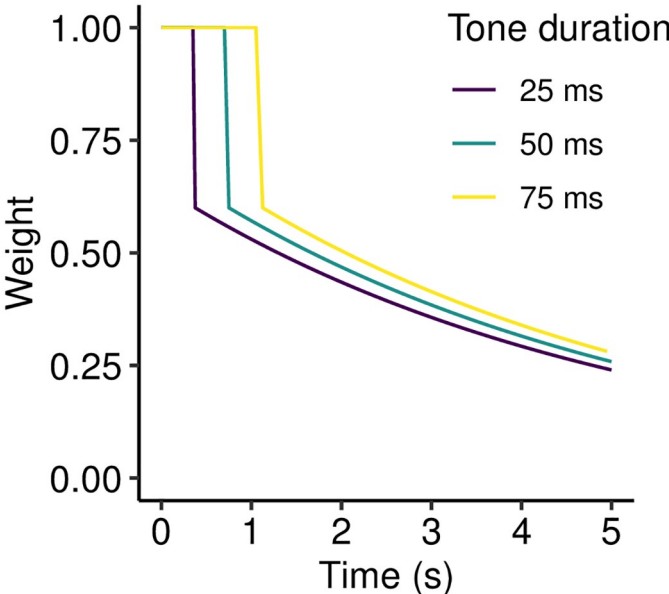

**Fig 7. Decay kernels employed in Experiment 3.** The temporal duration of the buffer corresponds to the buffer's informational capacity (15 tones) multiplied by the tone duration.

25-ms tones, 0.75 s for 50-ms tones, and 1.125 s for 75-ms tones. While the $n$-gram observation remains within this buffer, its weight is $w_0 = 1.0$; once the memory exits the buffer its weight drops to $w_1 = 0.6$, and thereafter decays exponentially to $w_\infty = 0$ with a half life of $t_{0.5} = 3.5$ s. The precise parameters of this decay kernel come from manual optimization to the behavioral data, and it should be noted that these parameters may well be task-dependent; nonetheless, we will show that each qualitative component of the decay kernel seems to be necessary to explain the observed pattern of results.

Weight decay by itself is not sufficient to cause memory loss, because PPM computes its predictions using ratios of event counts, which are preserved under multiplicative weight decay. We therefore introduce stochastic noise to the memory retrieval component of the PPM model, meaning that weight decay reduces the signal-to-noise ratio, and thereby gradually eliminates the memory trace of the original observation. In our optimized model this noise component is implemented by sampling from a Gaussian with zero mean and standard deviation $\sigma_\epsilon = 0.8$, taking the absolute value, and adding this to the total weight over all observations for a given $n$-gram, this total weight being an arbitrarily large non-negative number. We avoid negative noise samples so as to avoid negative event counts, for which the PPM model is not well-defined.

Applied to an individual trial, the model returns the information content for each tone in the sequence, corresponding to the surprisingness of that tone in the context of the prior portion of the sequence (Fig 5). Following [3], we suppose that the listener identifies the transition from random to regular tone patterns by detecting the ensuing drop in information content. We model this process using a non-parametric change-detection algorithm that sequentially applies the Mann-Whitney test to identify changes in a time series' location while controlling the false positive rate to 1 in 10,000 observations [69].

All stimuli were statistically independent from one another, and so responses should not be materially affected by experiences on previous trials. For simplicity and computational efficiency, we therefore left the PPM-Decay model's long-term learning weight ($w_\infty$) fixed at zero, and reset the model's memory store between each trial.

We analyzed 6 different PPM-Decay configurations, aiming to understand how the model's different features contribute to task performance, and which are unnecessary for explaining the perceptual data. Specifically, we built the proposed model step-by-step from the original PPM model, first adding exponential decay, then adding retrieval noise, then adding the memory buffer. We tested three versions of the final model with different buffer capacities: 5 items, 10 items, and 15 items. We manually optimized each model configuration to align mean participant response times to mean model response times, producing the parameter sets listed in Table 1.

**Table 1. Optimized model parameters for Experiment 3.**

| Model | $m_{max}$ | $t_b$ | $n_b$ | $w_0$ | $w_1$ | $t_{0.5}$ | $w_\infty$ | $\sigma_\epsilon$ |
|---|---|---|---|---|---|---|---|---|
| Original PPM | 4 | 0 | 0 | 0.0 | 1.00 | $\infty$ | 0 | 0.00 |
| + Exponential decay | 4 | 0 | 0 | 0.0 | 1.00 | **0.26** | 0 | 0.00 |
| + Retrieval noise | 4 | 0 | 0 | 0.0 | **0.65** | **1.65** | 0 | **0.50** |
| + 5-item buffer | 4 | $\infty$ | **5** | **2.0** | **0.70** | **1.45** | 0 | 0.50 |
| + 10-item buffer | 4 | $\infty$ | **10** | **1.5** | **0.40** | **1.90** | 0 | **0.35** |
| + 15-item buffer | 4 | $\infty$ | **15** | **1.0** | **0.60** | **3.50** | 0 | **0.80** |

Bold denotes parameters manipulated from the previous step. $m_{max}$ is the model's Markov order bound. $t_b$ is the temporal buffer capacity, $n_b$ the itemwise buffer capacity. $w_0$ is the buffer weight, $w_1$ is the initial post-buffer weight, and $w_\infty$ is the asymptotic post-buffer weight. $\sigma_\epsilon$ is the scale parameter for the retrieval noise distribution.

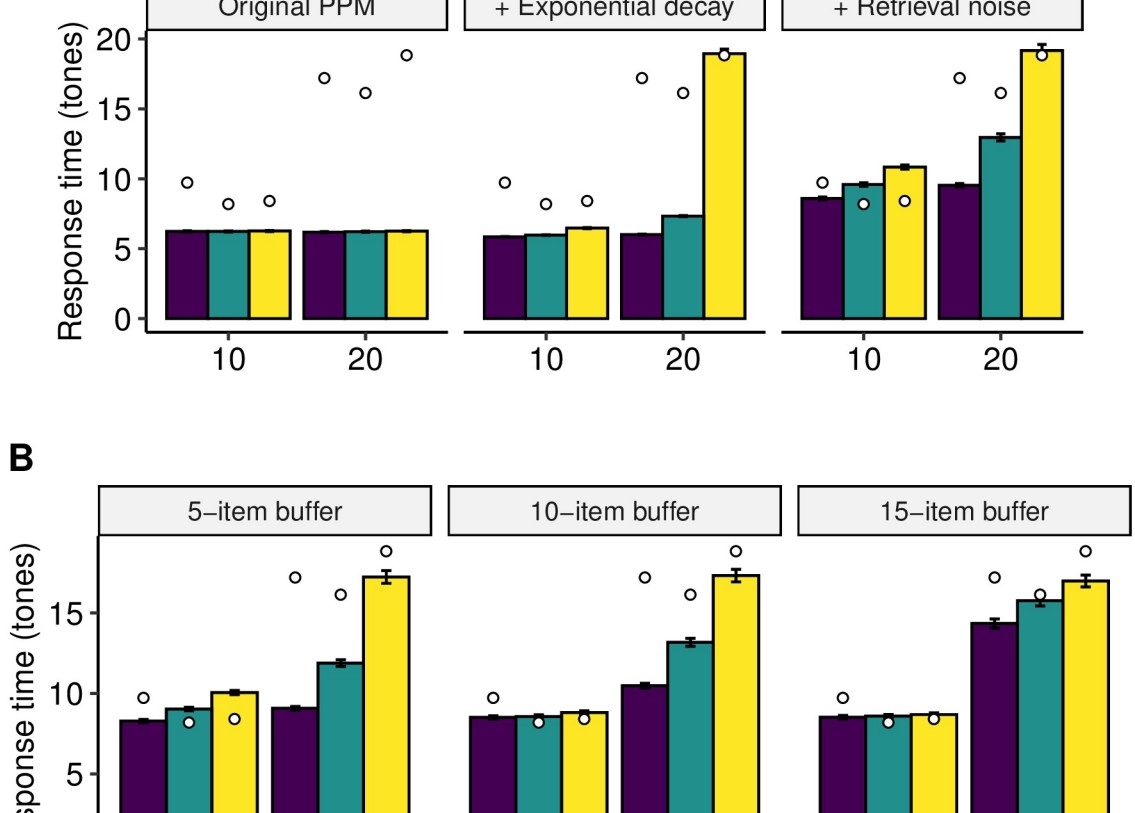

**Fig 8. Modeling participant data in Experiment 3.** Participant data (mean response times) are plotted as white circles, whereas different model configurations (mean simulated response times) are plotted as solid bars. Error bars denote 95% confidence intervals computed using the central limit theorem. **A)** Progressively adding exponential weight decay and retrieval noise to the original PPM model. **B)** Progressively adding longer buffers to the PPM-Decay model.

**Original PPM.** As expected, the original PPM model proved not to be sensitive to tone length or to alphabet size (Fig 8A). Furthermore, the model systematically outperformed the participants, with an average reaction time of 6.23 tones compared to the participants' mean reaction time of 12.90.

**Adding exponential decay.** Here we add time-based exponential decay, as in Experiments 1 and 2. One might expect this feature to induce a negative relationship between pattern-detection performance and cycle length. We do observe such an effect, but only with a very fast memory-decay rate (half life = 0.26 s; Fig 8A). This robustness of models without retrieval noise to memory decay can be rationalized by observing that, even as absolute weights of memory traces decrease with memory decay, the important information, namely the ratios of these weights, remains more or less preserved, and so the pattern-detection algorithm continues to perform well. Further to this, the model is problematic in that it

substantially outperforms participants in the 10-tone conditions, and exhibits no clear discontinuity in performance between the 10-tone conditions and the 20-tone conditions.

**Adding retrieval noise.** Retrieval noise increases the model's sensitivity to memory decay, and means that the drop in performance from the shortest cycles (10 tones, 25 ms/tone) to the longest cycles (20 tones, 75 ms/tone) can be replicated with a more plausible half-life of 1.65 s (Fig 8A). However, the model still fails to capture the discontinuity in reaction times between 10-tone and 20-tone conditions, especially with tone lengths of 25 and 50 ms.

**Adding the memory buffer.** We anticipated that a buffer with an informational capacity limit between 10 tones and 20 tones should be able to replicate the behavioral discontinuity between 10-tone and 20-tone conditions. The 10-tone cycles should largely fit in such a buffer, resulting in near-ceiling performance in the 10-tone conditions; conversely, the 20-tone cycles should be too big for the buffer, resulting in performance deterioration. Fig 8B shows that such an effect does indeed take place with a 15-tone buffer. In contrast, shorter buffers (5 tones, 10 tones) do not elicit this clear discontinuity between 10-tone and 20-tone conditions. The resulting model also replicates the insensitivity to tone duration in the 10-tone conditions, and the adverse effect of increasing tone duration to 75 ms in the 20-tone condition that was hinted at in the behavioral data. It therefore seems clear that a PPM-Decay model with a finite-capacity buffer can explain the main patterns of reaction times observed in this experiment, in contrast to the original PPM model.

Finally, we performed a simple sensitivity analysis to understand how much the model's fit depended on the precise values of the model parameters. We took the five optimized model parameters (itemwise buffer capacity, buffer weight, half life, initial post-buffer weight, and retrieval noise), and represented each as a Gaussian with mean equal to its optimized value and with standard deviation equal to 15% of the mean. We then sampled 100 parameter vectors from these Gaussians, reran the simulation with each of these parameter vectors, and compared the resulting model fits to that of the original parameter vector. We quantified model fit in terms of the consistency between mean model reaction times and mean participant reaction times, with reaction times measured in tones and aggregated by condition, and with consistency operationalized by two coefficients: the intraclass correlation coefficient (one-way, single measurement, absolute agreement), and the Spearman correlation coefficient. The former assesses how successfully the model predicts absolute reaction times, whereas the latter assesses how successfully the model predicts the order of reaction times. We found a mean intraclass correlation coefficient of .77 for the resampled parameters, compared to a coefficient of .96 for the original parameters, indicating that absolute model fit is moderately sensitive to the choice of parameters. In contrast, the mean Spearman correlation coefficient for the resampled parameters was essentially identical to that of the original parameters ($\rho$s = .83, .83), indicating that the qualitative trends in the model predictions are comparatively robust to local changes in the parameters.

Here we supposed that our observed trends in reaction times reflect auditory memory decay. It is worth considering alternative explanations for these trends. One possibility is that the trends reflect interference in auditory memory rather than memory decay *per se* [70]. This would be an important mechanistic distinction to make, but would not change our main conclusions about the importance of memory limitations for auditory prediction and the presence of a finite-capacity auditory memory buffer. A second possibility is that the reaction-time trends were not driven by memory decay, but were instead driven by older memories taking longer to retrieve from memory than newer memories. Likewise, this would be an important mechanistic distinction, but would not change our main conclusions. A third possibility is that the reaction-time trends were driven by sensory processing limitations. However, such limitations would predict that the faster tones be processed less effectively, meaning that more tones

would be required for change detection in fast-tone conditions than in slow-tone conditions, which is not what we observe. Memory dynamics therefore still seem to provide the most plausible explanation for these effects.

## Discussion

PPM is a powerful sequence prediction algorithm that has proved well-suited to modeling the cognitive processing of auditory sequences [3, 5, 19–22]. In these contexts, PPM has traditionally been interpreted as an ideal observer, simulating an (approximately) optimal strategy for predicting upcoming auditory events on the basis of learned statistics. This modeling strategy has proved very useful for elucidating the role of statistical cognition in auditory perception [3, 16].

Here we introduced a customizable decay kernel to PPM, which downweights historic observations as time passes and subsequent observations are registered in memory. This decay kernel is useful for two primary reasons. First, it makes PPM a better approximation to an ideal observer when the underlying sequence statistics change over time, as is common in many real-world listening contexts. Second, it allows the model to capture the multi-stage nature of human auditory memory, with its corresponding capacity limitations and temporal profiles.

We applied this new PPM-Decay model in three experiments. The first experiment analyzed sequences generated from a statistical model whose underlying parameters evolved over time, and verified that PPM-Decay better approximates an ideal observer than PPM when applied to such sequences. The second experiment simulated a musically naive listener who gradually learns to predict chord progressions through exposure to compositions from three musical styles: popular music, jazz music, and chorale harmonizations by J. S. Bach. Again, we found that PPM-Decay better approximated an ideal observer than the original PPM model. The ideal model configuration incorporated a recency effect, reflecting how the underlying statistics of the chord progressions differ between compositions, and evolve during the course of individual compositions. However, the model's decay kernel also incorporated a positive asymptote, allowing the model to develop long-term knowledge of certain statistical regularities that are shared between different compositions from the same musical style.

The third experiment revisited an auditory detection paradigm from [3], where participants had to detect transitions between random and regular sections in tone sequences that varied in alphabet size and tone length. The original authors found tentative evidence for auditory pattern detection being constrained by the capacity limitations of echoic memory, but were unable to determine whether these results reflected temporal limitations (e.g. echoic memory only spans two seconds) or informational limitations (e.g. echoic memory can only hold up to 15 tones). We conducted a new behavioral experiment using stimuli designed to distinguish these two possibilities, by varying tone duration and the number of tones in the regular patterns independently. The resulting data implied that human performance stayed constant as long as the relevant auditory input could fit within a buffer of limited itemwise capacity. We formalized this explanation computationally with our PPM-Decay model, and showed that the model could successfully reproduce the observed behavioral data, in contrast to simpler model variants such as the original PPM model [3, 18, 43] or a PPM model with solely exponential memory decay. Now, this particular experiment only provided a limited amount of data with which to evaluate this relatively complex model, and it is difficult to dismiss the possibility of alternative model configurations predicting the data equally well. Nonetheless, our model provides a plausible hypothesis for how the observed response patterns came about, and our

experiment provides a useful starting point for developing computational models that better resemble human auditory prediction.

It is interesting to note how memory decay can behave both as a cognitive limitation (by constraining the amount of information that can be retained by the observer) and an adaptive strategy for predicting sequences whose statistical structure changes over time (by prioritizing recent and hence more informative statistics). In line with these observations, it seems likely that the dynamics of memory decay in human listeners reflect both the constraints of limited cognitive resources and the mutability of natural auditory environments.

We anticipate that this PPM-Decay model should prove useful for other applications in auditory modeling. The combination of the statistical power of PPM and the flexible decay kernel makes the model well-suited to simulating online auditory statistical learning under memory constraints and in changing statistical environments. A particularly relevant application domain is music cognition, which has already made significant use of PPM models without decay kernels [19–22, 48, 71]. Incorporating decay kernels into these models should be useful for capturing how recency effects and memory limitations influence the probabilistic processing of musical structure. However, the PPM-Decay algorithm itself is relatively domain-agnostic, and should be applicable to any sequential domain where observations can be approximated as discrete symbols drawn from a finite alphabet. We anticipate that our publicly available R package "ppm" should prove useful for supporting such work (https://github.com/pmcharrison/ppm).

An important avenue for future work is to improve our understanding of the ideal decay kernels for different modeling applications. When optimizing a decay kernel for predictive performance on a corpus of sequences, we learn about the statistical structure of that corpus, specifically the sense in which historical events of different vintages contribute useful information about upcoming events. Such analyses are particularly relevant to computational musicology, where a common goal is to quantify statistical processes underlying music composition. When optimizing a decay kernel to reproduce human performance, we learn about the predictive strategies actually used by humans, and the sense in which they may be constrained by cognitive limitations. The optimized decay kernel from Experiment 3 provides an initial model that seems to account well for the behavioral data collected here, but further empirical work is required to constrain the details of this model and to establish its generalizability to different experimental contexts.

A primary limitation of the PPM and PPM-Decay models is that they operate over discrete representations, and do not model the process by which these discrete representations are extracted from the auditory signal. This simplification is convenient when modeling systems such as music and language, which are often well-suited to symbolic expression, but it is problematic when modeling continuous stimulus spaces. One solution to this problem is to adopt continuous-input models [28, 72], where discretization plays no part; however, such models typically struggle to capture the kinds of structural dependencies common in music and language, and do not reflect the apparent importance of categorical perception in human auditory perception [73]. One alternative way forward might be to prefix the PPM-Decay model with an unsupervised discretization algorithm, such as $k$-means clustering [74].

The PPM-Decay algorithm can become computationally expensive with long input sequences. In the naive implementation, the algorithm must store an explicit record of each $n$-gram observation as it occurs, meaning that the time and space complexity for generating a predictive distribution is linear in the length of the training sequence. However, particular families of decay kernels can support more efficient implementations. For example, a decay kernel comprising the sum of $N$ exponential functions can be implemented as a set of $N$ counters for each $n$-gram, each of which is incremented upon observing the respective $n$-gram,

and each of which is decremented by a fixed ratio at each timestep. This implementation has bounded time and space complexity with regard to the length of the training sequence. Such approaches should be useful for speeding the application of the PPM-Decay model to large datasets, and for improving its biological plausibility.

The PPM and PPM-Decay models assume that listeners process auditory stimuli by computing transition probabilities from memories of *n*-gram observations. While *n*-gram models seem to provide a good account of auditory processing [3, 16], they may not be sufficient to explain all aspects of auditory learning. For example, *n*-gram models struggle to explain how listeners can (albeit with some difficulty) learn non-adjacent dependencies [75, 76] or recursive grammatical structures [10, 77]. Some of these phenomena might be explained by incorporating further modifications to the memory model; for example, non-adjacent dependencies could be learned by combining *n*-gram modeling with the abstraction method of [78]. Other phenomena, such as the acquisition of recursive grammars, might only be explained by alternative modeling approaches. This remains a challenge for future research.

The current PPM-Decay model uses memory decay to cope with changes in sequence statistics. This technique works well if sequence statistics are always changing to new, unfamiliar values; however, in the natural world it is quite possible that historic sequence statistics end up returning in the future. In such scenarios, it would be desirable to have retained a memory of these historic statistics, and to reactivate this memory once it becomes relevant again. It seems plausible that humans do something like this; for example, individuals who learn multiple languages seem able to learn separate models for each language, and switch between these models according to context. Something similar may happen when individuals become familiar with multiple musical styles [79]. In the domain of natural language processing, various techniques have been developed that implement such a model switching process, with input sequences being allocated to models using either prespecified assignments or unsupervised clustering [80–83]. It would be worthwhile to incorporate similar techniques into the PPM-Decay model.

The memory-decay function in the PPM-Decay model currently only depends on the time-points of event observations. However, one might expect memory retention to be influenced by various other phenomena, such as attention [84, 85], or sleep consolidation [86–88]. Though such phenomena are not currently captured by the PPM-Decay model, it should be possible to address them through various modifications to the memory-decay function without having to modify the wider modeling framework.

The informational buffer capacity of the PPM-Decay model is currently defined as a fixed number of items, implying that each auditory event occupies an identically sized portion of auditory memory. However, it is possible that the auditory memory buffer is not restricted so much by the number of events but rather by the amount of information that these events represent. According to this proposition, highly expected or repetitive events would occupy less of the buffer, hence increasing its effective itemwise capacity [63, 89]. It would be interesting to test a variant of the PPM-Decay model that implements such a buffer, perhaps quantifying the information carried by each event as its negative log probability according to the predictive model, after [90].

The PPM-Decay model addresses how listeners generate predictions based on a learned generative model of sequential structure: the resulting predictions may be termed 'schematic expectations'. However, given sufficient exposure to particular sequences, listeners may memorize the sequences themselves, allowing them to produce what may be termed 'veridical expectations'. While schematic expectations are relevant for individuals learning new languages and new musical styles, veridical expectations are relevant for individuals memorizing passages of literature or particular pieces of music. It seems that these two kinds of expectations may constitute distinct psychological mechanisms [91, 92]; future work is required to understand how well the PPM-Decay model can extend past schematic expectations to veridical expectations.

We modeled musical corpora using PPM-Decay with an exponentially decaying memory kernel. However, it has been argued that sequential dependencies in many natural sequences (e.g. language, birdsong) have power-law decay, not exponential decay [93, 94]. To the extent that music is characterized by syntactic expressivity greater than that of a probabilistic regular grammar [95], we would also expect it to exhibit power-law dependencies [96]. It has been argued that human memory also exhibits power-law decay, making it ideally suited to processing natural language [94], though see [97–99] for reasons to be cautious. A natural follow-up would therefore be to compare PPM-Decay models with exponential decay to equivalent models with power-law decay; if natural sequences and human memory are better characterized by power-law decay, we would expect the latter models to generate more accurate sequence predictions and better simulations of participant behavior.

We chose to focus on extending PPM for the present work because PPM already plays an important role in the predictive processing literature. However, there are many other algorithms that could theoretically be used in its place. When choosing a potential replacement for PPM, it is important to bear three principles in mind: a) how well the algorithm approximates an ideal observer, as quantified by corpus modeling; b) how plausible the algorithm would be as a cognitive model; c) how easily the algorithm can be customized to implement cognitively motivated features. One prominent competitor from the data-compression literature is the Lempel-Ziv algorithm [100], an example of a dictionary-coding model that can be formulated as an incremental sequence prediction model like PPM [101–103]. Formulated this way, Lempel-Ziv has a similar form to PPM, but one that trades PPM's complex blending strategies for increased computational efficiency. Consequently, PPM systematically outperforms Lempel-Ziv algorithms on data compression benchmarks [103–105]; we therefore prefer PPM for our purposes, as it provides a better approximation to an ideal observer. A second prominent competitor from the data-compression literature is the PAQ family of algorithms, which takes the $n$-gram blending principle of PPM as a starting point and adds many additional features, such as neural networks for combining sub-models, two-dimensional contexts for modeling spreadsheets, and specialized models for BMP, TIFF, and JPEG image formats [106]. While the PAQ algorithms outperform PPM on standard data compression benchmarks [39], their specialization to computer file formats makes them implausible as cognitive models. A third potential competitor from the machine-learning literature is the recurrent neural network, a model that (given sufficiently many parameters and appropriate training) has a powerful universal approximator property [107]. However, such networks typically require a lot of training data to reach high performance levels, and it is difficult to see how they could incorporate fully customizable memory decay functions (though some kinds of memory decay could be achieved through a combination of online learning and regularization). In conclusion, we chose to use PPM as it provides an effective balance of predictive power, cognitive plausibility, and customizability that makes it well-suited to modeling auditory predictive processing; nonetheless, future research might profitably consider potential alternative models, and seek to collect cognitive data that can help choose between such models.

Several alternative cognitive models of sequence prediction have explicitly Bayesian formulations [29, 30, 72]. This approach is appealing because it formally motivates the predictive algorithm from a set of assumptions about the underlying sequence statistics. Such approaches can also be applied to mixed-order Markov models such as PPM, but typically they come with substantially increased computational complexity [108], which may prove impractical for many cognitive modeling applications. Nonetheless, it would be worth examining how the present approaches might be motivated as computationally efficient approximations to Bayes-optimal models.

## Methods

### Ethics statement

The present research was approved by the research ethics committee of University College London (Project ID Number: 1490/009). Written informed consent was obtained from each participant.

### Model

Our PPM-Decay model embodies a predictive processing account of auditory regularity detection. It supposes that listeners acquire an internal model of incoming sounds through automatic processes of statistical learning, and use this model to generate prospective predictions for upcoming auditory events. The model derives from the PPM algorithm [17, 43], but adds three psychological principles:

a) The memory salience of a given observation decays as a function of the timepoints of subsequently observed events and the timepoint of memory retrieval.

b) There exists some noise, or uncertainty, in memory retrieval.

c) A limited-capacity memory buffer constrains learning and prediction. Contiguous events (*n*-grams) must fit into this buffer to be internalized or to contribute to prediction generation.

Each of these three features can be enabled or disabled in isolation. In ideal-observer analyses, such as Experiments 1 and 2, it often makes sense to omit features b) and c), because they correspond to cognitive constraints that typically impair prediction. Here we therefore omit these two features for the ideal-observer analyses (Experiments 1 and 2), but retain them for the behavioral analyses in Experiment 3.

Many variants of PPM exist in the literature [17, 40, 43, 109]. Our formulation incorporates the interpolated smoothing technique of [43], but avoids techniques such as Kneser-Ney smoothing, exclusion, update exclusion, and state selection, because it is not obvious how to generalize these techniques to decay-based models where *n*-gram counts can fall between zero and one, and because we do not yet have compelling evidence that human cognition employs analogous methods. Nonetheless, an interesting topic for future work is to explore the cognitive relevance and potential computational implementations of these techniques.

**Domain.** The model assumes that the auditory input can be represented as a sequence of symbols drawn from a discrete alphabet; the cognitive processes involved in developing this discrete representation are not addressed here. Let $\mathcal{A}$ denote the discrete alphabet, let () denote an empty sequence, and let $e_1^N = (e_1, e_1, \ldots, e_N)$ denote a sequence of $N$ symbols, where $e_i \in \mathcal{A}$ is the $i$th symbol in the sequence, and $e_i^j$ is defined as

$$e_i^j = \begin{cases} (e_i, e_{i+1}, \ldots, e_j) & \text{if } i \leq j, \\ () & \text{otherwise.} \end{cases}$$

We suppose that this sequence is presented over time, and denote the timepoint of the $i$th symbol as $\tau_i$.

Now suppose that $E_1^N$ is a random variable corresponding to a sequence of length $N$. We consider an observer predicting each symbol of $E_1^n$ based on the previously observed symbols. This corresponds to the probability distribution $P(E_i = e_i | E_1^{i-1} = e_1^{i-1})$, which we will abbreviate as $P(e_i | e_1^{i-1})$. The model is tasked with estimating this conditional probability distribution.

**Learning.** The model learns by counting occurrences of different sequences of length $n$ termed *n-grams* ($n \in \mathbb{N}^+$), where $n$ is termed the *n-gram order*. As in PPM, the model counts

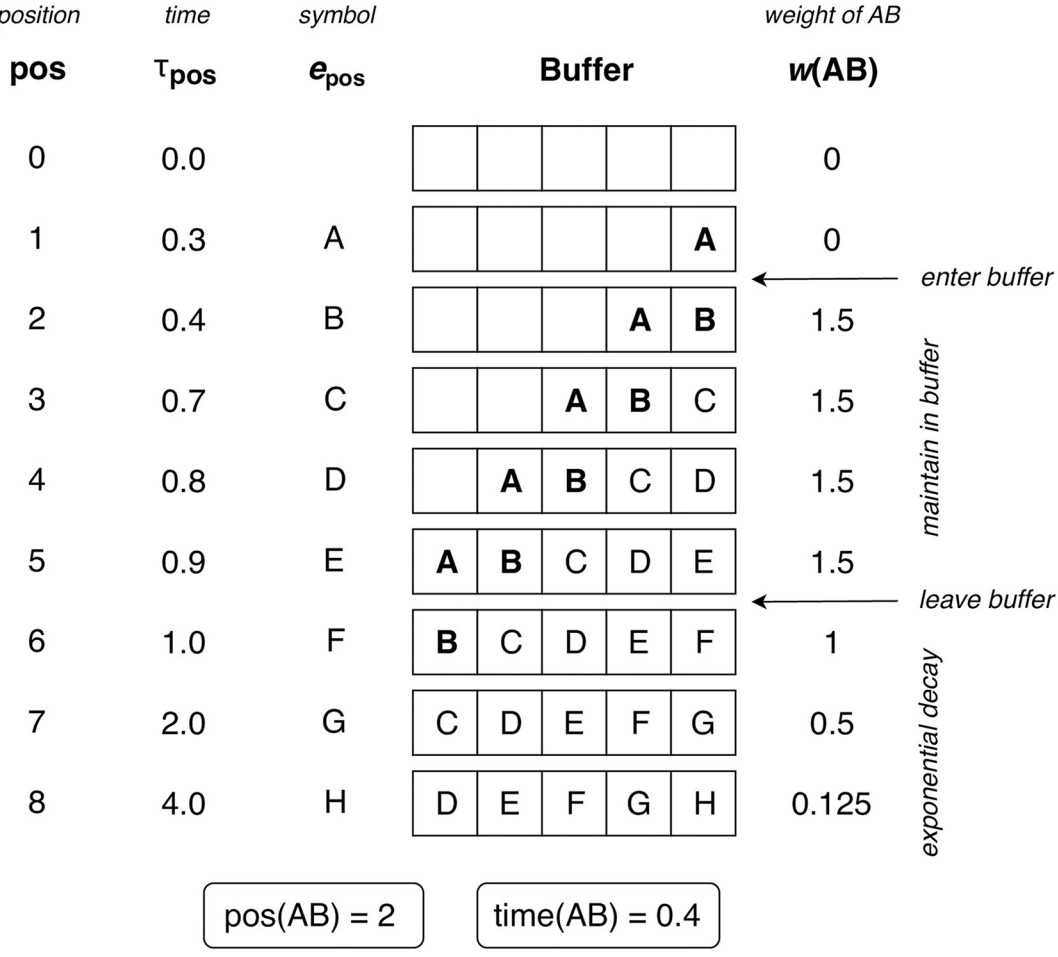

**Fig 9. Schematic figure of accumulating observations within a memory buffer.** Weights for the $n$-gram "AB" are displayed as a function of time, assuming an itemwise buffer capacity ($n_b$) of 5, a buffer weight ($w_0$) of 1.5, an initial post-buffer weight ($w_1$) of 1, a half life ($t_{0.5}$) of 1 second, and an asymptotic post-buffer weight ($w_\infty$) of 0.

$n$-grams for all $n \leq n_{max}$ ($n_{max} \in \mathbb{N}^+$), where $n_{max}$ is the *n-gram order bound*. A three-symbol sequence ($e_1, e_2, e_3$) contains six $n$-grams: ($e_1$), ($e_2$), ($e_3$), ($e_1, e_2$), ($e_2, e_3$), and ($e_1, e_2, e_3$).

We suppose that $n$-grams are extracted from a finite-capacity buffer (Fig 9). Successive symbols enter and leave this buffer in a first-in first-out arrangement, so that the buffer represents a sliding window over the input sequence. The buffer has two capacity limitations: *itemwise capacity* and *temporal capacity*. The itemwise capacity, $n_b$, determines the maximum number of symbols stored by the buffer; the temporal capacity, $t_b$, determines the maximum amount of time that a given symbol can remain in the buffer before expiry. Generally speaking, itemwise capacity will be the limiting factor at fast presentation rates, whereas temporal capacity will be the limiting factor at slow presentation rates. As $n$-grams may only be extracted if they fit completely within the buffer, these capacities bound the order of extracted $n$-grams. Correspondingly, we constrain $n_{max}$ (the $n$-gram order bound) not to exceed $n_b$ (the itemwise buffer capacity).

In PPM, $n$-gram observations are recorded by incrementing a counter. Our PPM-Decay model also stores the ordinal position within the input sequence when the observation occurred; this is necessary for simulating the temporal dynamics of auditory memory. For

each $n$-gram $x$, we define count($x$) as the total number of observations of $x$, and pos($x$) as a list of ordinal positions in the input sequence when these observations occurred, defined with respect to the final symbol in the $n$-gram. pos($x$) is initialized as an empty list; each time a new $n$-gram $x$ is observed, the respective ordinal position is appended to the list. count($x$) is then represented implicitly as the length of pos($x$).

The input sequence is processed one symbol at a time, from beginning to end. Observing the $i$th symbol, $e_i$, yields up to $n_{max}$ $n$-gram observations, corresponding to all $n$-grams in the buffer that terminate with the most recent symbol: $\{e^i_{i-n+1} : n \leq \min(i, n_{max})\}$. If the buffer component of the model is enabled, an $n$-gram observation will only be recorded if it fits completely within the itemwise and temporal capacities of the buffer; the former constraint is ensured by the constraint that $n_{max} \leq n_b$, but the latter must be checked by comparing the current timepoint (corresponding to the final symbol in the $n$-gram) with the timepoint of the first symbol of the $n$-gram. If the current ordinal position is written pos$_{end}$, and the $n$-gram length is written size($x$), then the necessary and sufficient condition for $n$-gram storage is

$$\text{time}_{end} - \text{time}_{start} \leq t_b$$

where

$$
\begin{aligned}
\text{time}_{end} &= \tau_{\text{pos}_{end}} \\
\text{time}_{start} &= \tau_{\text{pos}_{start}} \\
\text{pos}_{start} &= \text{pos}_{end} - \text{size}(x) + 1,
\end{aligned}
$$

$\tau_i$ is the $i$th timepoint in the input sequence, and $t_b$ is the temporal buffer capacity, as before. Table 2 describes the information potentially learned from training on the sequence $(a, b, a)$.

**Memory decay.** In the original PPM algorithm, the influence of a given $n$-gram observation is not affected by the passage of time or the encoding of subsequent observations. This contrasts with the way in which human observers preferentially weight recent observations over historic observations [26, 27, 29–31, 44, 110]. This inability to capture recency effects limits the validity of PPM as a cognitive model.

Here we address this problem. We suppose that the influence, or *weight*, of a given $n$-gram observation varies as a function both of the current timepoint and the timepoints of the symbols that have since been observed. This weight decay function represents the following hypotheses about auditory memory:

1. Each $n$-gram observation begins in the memory buffer (Fig 9). Within this buffer, observations do not experience weight decay.

2. Upon leaving the buffer, observations enter a secondary memory store. This transition is accompanied by an immediate drop in weight.

**Table 2. $n$-grams learned from training on the sequence a, b, a.**

| $x$ | count($x$) | pos($x$) |
|---|---|---|
| $(a)$ | 2 | 1, 3 |
| $(b)$ | 1 | 2 |
| $(a, b)$ | 1 | 2 |
| $(b, a)$ | 1 | 3 |
| $(a, b, a)$ | 1 | 3 |

3. While in the secondary memory store, observations experience continuous weight decay over time, potentially to a non-zero asymptote.

These hypotheses must be considered tentative, given the scarcity of empirical evidence directly relating memory constraints to auditory prediction. However, the notion of a short-lived memory buffer is consistent with pre-existing concepts of auditory sensory memory [23–25], and the continuous-decay phenomenon is consistent with well-established recency effects in statistical learning [26, 27, 29–31, 44, 110].

We formalize these ideas as follows. For readability, we write $pos(x, i)$ for the $i$th element of $pos(x)$, corresponding to the ordinal position of the $i$th observation of $n$-gram $x$ within the input sequence, defined with respect to the final symbol of the $n$-gram. Similarly, we write $time(x, i)$ as an abbreviation of $\tau_{pos(x, i)}$, the timepoint of the $i$th observation of $n$-gram $x$. We then define $w(x, i, t)$ as the weight for the $i$th observation of $n$-gram $x$ for an observer situated at time $t$:

$$w(x, i, t) = \begin{cases} w_0 & \text{if } t \leq \text{time}_{\text{expire}}(x, i), \\ w_\infty + (w_1 - w_\infty)f(t - \text{time}_{\text{expire}}(x, i)) & \text{otherwise.} \end{cases}$$

Here $w_0$ is the *buffer weight*, $w_1$ is the *initial post-buffer weight*, and $w_\infty$ is the *asymptotic post-buffer weight* ($w_0 \geq w_1 \geq w_\infty \geq 0$). The function $f$ defines an exponential decay with half-life equal to $t_{0.5}$, with $t_{0.5} > 0$:

$$\begin{aligned} f(t) &= \exp(-\lambda t) \\ \lambda &= \log(2)/t_{0.5}. \end{aligned}$$

$\text{time}_{\text{expire}}(x, i)$ denotes the timepoint at which the $i$th observation of $n$-gram $x$ expires from the buffer, computed as the earliest point when either the temporal capacity or the itemwise capacity expires. The temporal capacity expires when $t_b$ seconds have elapsed since the first symbol in the $n$-gram, whereas the itemwise capacity expires when $n_b$ symbols have been observed since the first symbol in the $n$-gram:

$$\begin{aligned} \text{time}_{\text{expire}}(x, i) &= \min\left(\text{time}_{\text{temporal expiry}}(x, i), \text{time}_{\text{itemwise expiry}}(x, i)\right) \\ \text{time}_{\text{temporal expiry}}(x, i) &= \text{time}_{\text{begin}}(x, i) + t_b \\ \text{time}_{\text{begin}}(x, i) &= \tau_{\text{pos}_{\text{begin}}(x,i)} \\ \text{pos}_{\text{begin}}(x, i) &= \text{pos}(x, i) - \text{size}(x) + 1 \\ \text{time}_{\text{itemwise expiry}}(x, i) &= \begin{cases} \infty & \text{if } \text{pos}_{\text{itemwise expiry}}(x, i) > N, \\ \tau_{\text{pos}_{\text{itemwise expiry}}(x,i)} & \text{otherwise,} \end{cases} \\ \text{pos}_{\text{itemwise expiry}}(x, i) &= \text{pos}_{\text{begin}}(x, i) + n_b. \end{aligned}$$

An illustrative memory-decay profile is shown in Fig 10.

Memory traces accumulate over repeated observations of the same $n$-gram. We define $W(x, t)$, the accumulated weight for an $n$-gram $x$, as

$$W(x, t) = \sum_{i:1 \leq i \leq \text{count}(x)} w(x, i, t).$$

As currently specified, memory decay does not necessarily cause forgetting, because the same information may be preserved in the ratios of $n$-gram weights even as the absolute values of the weights shrink. For example, consider a pair of $n$-grams $AB$ and $AC$ with weights 4 and

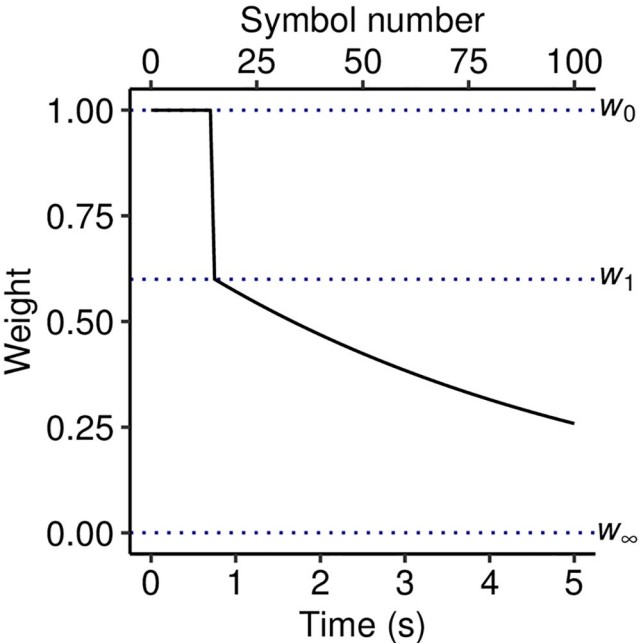

**Fig 10. Illustrative weight decay profile.** This figure plots the weight of an $n$-gram of length one as a function of relative observer position, assuming that new symbols continue to be presented every 0.05 seconds. Model parameters are set to $t_b = 2$, $n_b = 15$, $w_0 = 1.0$, $t_{0.5} = 3.5$, $w_1 = 0.6$, and $w_\infty = 0$, as optimized in Experiment 3.

1 respectively, both undergoing exponential decay to an asymptotic weight of 0. From these $n$-gram weights, the model can estimate the probability that $B$ follows $A$ as $p(B|A) = 4/(4 + 1) = 0.8$. After one half-life, the new counts are 2 and 0.5 respectively, but the maximum-likelihood estimate remains unchanged: $p(B|A) = 2/(2 + 0.5) = 0.8$.

A better account of forgetting can be achieved by supposing that memory traces must compete with noise factors introduced by imperfections in auditory memory; in this case, shrinking the absolute values of $n$-gram weights decreases their signal-to-noise ratio and hence induces forgetting. Here we model imperfections in memory retrieval by adding truncated Gaussian noise to the retrieved weights:

$$W^*(x, t) = W(x, t) + \max(0, \epsilon) \tag{1}$$

where $W^*(x, t)$ is the retrieved weight of $n$-gram $x$ at time $t$, and $\epsilon \sim N(0, \sigma_\epsilon^2)$ represents Gaussian noise uncorrelated across $n$-grams or timepoints. Setting $\sigma_\epsilon^2$ to zero disables the noise component of the model.

**Prediction.** Traditionally, a maximum-likelihood $n$-gram model estimates the probability of symbol $e_i$ given context $e_1^{i-1}$ by taking all $n$-grams beginning with $e_{i-n+1}^{i-1}$ and finding the proportion that continued with $e_i$. For $n \leq i$:

$$P(e_i|e_1^{i-1}) \approx \hat{P}_n(e_i|e_1^{i-1}) = \begin{cases} 1/|\mathcal{A}| & C_n(e_1^{i-1}) = 0, \\ c(e_{i-n+1}^i)/C_n(e_1^{i-1}) & \text{otherwise.} \end{cases}$$

$$C_n(e_1^{i-1}) = \sum_{x \in \mathcal{A}} c(e_{i-n+1}^{i-1} :: x)$$

where $\hat{P}_n$ denotes an $n$-gram probability estimator of order $n$, $c(e_i^j)$ is the number of times $n$-gram $c(e_i^j)$ occurred in the training set, and $e_i^j :: x$ denotes the concatenation of sequence $e_i^j$ and

symbol $x$. The $n$-gram model predicts from the previous $n - 1$ symbols, and therefore constitutes an $(n - 1)$th-order Markov model. Note that the estimator defaults to a uniform distribution if $C_n(e_1^{i-1}) = 0$, when the context has never been seen before. Note also that the predictive context of a 1-gram model is the empty sequence $e_i^{i-1} = ()$.

To incorporate memory decay into a maximum-likelihood $n$-gram model, we replace the count function $c$ with the retrieval weight function $W^*$. For $n \leq i$:

$$P(e_i|e_1^{i-1}) \approx \hat{P}_n(e_i|e_1^{i-1}) = \begin{cases} 1/|\mathcal{A}| & T_n(e_1^{i-1}) = 0, \\ W^*(e_{i-n+1}^i, \text{time}(e_i))/T_n(e_1^{i-1}) & \text{otherwise.} \end{cases}$$

$$T_n(e_1^{i-1}) = \sum_{x \in \mathcal{A}} W^*(e_{i-n+1}^{i-1} :: x, \text{time}(e_i))$$

This decay-based model degenerates to the original maximum-likelihood model when $w_0 = 1$, $t_b \to \infty$, $n_b \to \infty$, $\sigma_\epsilon = 0$ (i.e. an infinite-length memory buffer with unit weight and no retrieval noise).

High-order $n$-gram models take into account more context when generating their predictions, and are hence capable of greater predictive power; however, this comes at the expense of greater tendency to overfit to training data. Conversely, low-order models are more robust to overfitting, but this comes at the expense of lower structural specificity. Smoothing techniques combine the benefits of both high-order and low-order models by merging $n$-gram models of different orders, with model weights varying according to the amount of training data. Here we use interpolated smoothing as introduced by [43, 104]. For $n \leq i$, the unnormalized interpolated $n$-gram estimator is recursively defined as a weighted sum of the $n$th-order maximum-likelihood estimator and the $(n - 1)$th-order interpolated estimator:

$$\hat{P}_n^*(e_i|e_1^{i-1}) = \begin{cases} 1/(|\mathcal{A}| + 1) & \text{if } n = 0, \\ \hat{P}_n(e_i|e_1^{i-1})a_n(e_1^{i-1}) + (1 - a_n(e_1^{i-1}))\hat{P}_{n-1}^*(e_i|e_1^{i-1}) & \text{otherwise,} \end{cases} \qquad (2)$$

where $\hat{P}_n^*$ is the $n$th-order unnormalized interpolated $n$-gram estimator, $\hat{P}_n$ is the $n$th-order maximum-likelihood estimator, $|\mathcal{A}|$ is the alphabet size, and $a_n$ is a function of the context sequence that determines how much weight to assign to $\hat{P}_n$, the maximum-likelihood $n$-gram estimator of order $n$.

The unnormalized interpolated estimator defines an improper probability distribution that does not necessarily sum to 1. We therefore define $\hat{P}_n^{**}$ as the normalized interpolated estimator:

$$\hat{P}_n^{**}(e_i|e_1^{i-1}) = \frac{\hat{P}_n^*(e_i|e_1^{i-1})}{\sum_{x \in \mathcal{A}} \hat{P}_n^*(x|e_1^{i-1})} \quad \text{for } n \leq i.$$

Note that the need for normalization can alternatively be avoided by redefining $\hat{P}_n^*(e_i|e_1^{i-1}) = 1/|\mathcal{A}|$ for $n = 0$ in Eq (2), meaning that the interpolated smoothing terminates with a proper probability distribution. However, we keep the original definition to preserve equivalence with [43] and [18].

The weighting function $a_n$ corresponds to the so-called "escape mechanism" of the original PPM algorithm. [111] review five different escape mechanisms, termed "A" [17], "B" [17], "C" [40], "D" [112], and "AX" [113] [43, 104], each corresponding to different weighting functions $a_n$. Of these, "C" tends to perform the best in data compression benchmarks [111]. However, methods "B", "C", "D", and "AX" do not generalize naturally to decay-based models; in

particular, it is difficult to ensure that the influence of an observation is a continuous function of its retrieved weight $w^*$. We therefore adopt mechanism "A".

In its original formulation, mechanism "A" gives the higher-order model a weight of $a_n = 1 - 1/(1 + T_n)$, where $T_n$ is the number of times the predictive context has been seen before (which can be interpreted as the observer's familiarity with the preceding sequence of $n - 1$ tokens). When the context has never been seen before, $T_n = 0$ and $a_n = 0$, and the estimator relies fully on the lowest-order model; as $T_n \rightarrow \infty$, $a_n \rightarrow 1$, and the estimator relies fully on the highest-order model. In the original PPM algorithm, the number of times that the predictive context has been seen before is equal to the sum of the weights (or counts) for each possible continuation:

$$T_n(e_1^{i-1}) = \sum_{x \in \mathcal{A}} W^*(e_{i-n+1}^{i-1} :: x, \text{time}(e_i)).$$

Introducing memory-decay reduces the weights for these prior observations, decreasing the model's effective experience, and preferentially weighting lower-order models, as might be expected. However, retrieval noise is problematic, because it positively biases the retrieved weights (see Eq (1)), causing the algorithm to overestimate its familiarity with its predictive context, and to overweight high-order predictive contexts as a result. We compensate for this by subtracting the expected value of the retrieval noise's contribution to $T_n$, which can be computed from standard results for the truncated normal distribution as $\sigma_\epsilon \sqrt{\frac{2}{\pi}}$, and truncating at zero:

$$T_n^*\left(e_1^{i-1}\right) = \max\left(0, T_n\left(e_1^{i-1}\right) - \sigma_\epsilon \sqrt{\frac{2}{\pi}}\right).$$

Putting this together, we have (for $i \geq n$):

$$a_n(e_1^{i-1}) = 1 - 1/(1 + T_n^*(e_1^{i-1}))$$

$$T_n^*(e_1^{i-1}) = \max\left(0, T_n\left(e_1^{i-1}\right) - \sigma_\epsilon \sqrt{\frac{2}{\pi}}\right)$$

$$T_n(e_1^{i-1}) = \sum_{x \in \mathcal{A}} W^*(e_{i-n+1}^{i-1} :: x, \text{time}(e_i)).$$

For its final output, the model selects the maximum-order available normalized interpolated estimator. The available orders are constrained by three factors:

1. The $n$-gram order bound: the model cannot predict using $n$-grams larger than $n_{max}$.

2. The sequence: the predictive context must fit within the observed sequence.

3. The buffer: the predictive context must fit within the buffer at the point when the incoming symbol is observed.

Putting this together, the selected $n$-gram order for generating predictions from a context of $e_1^{i-1}$ becomes:

$$\text{order}(e_1^{i-1}) = \max\left\{y \in \{0, 1, \ldots, n_{max}\} : y \leq i, \tau_i - \tau_{i-y+1} \leq t_b\right\}.$$

The final model output is then:

$$P(e_i|e_1^{i-1}) \approx \hat{P}^{**}_{\text{order}(e_1^{i-1})}(e_i|e_1^{i-1}).$$

The $n$-gram order bound, $n_{max}$, constrains the length of $n$-grams that are learned by the model. However, it is often more convenient to speak in terms of the model's *Markov order*, $m_{max}$, defined as the number of preceding symbols that contribute towards prediction generation. A single $n$-gram model generates predictions with a Markov order of $n - 1$; correspondingly, $m_{max} = n_{max} - 1$.

Fig 11 illustrates the interpolated smoothing mechanism. Here we imagine that a model with a Markov order bound of two processes the sequence "abracadabra", one letter at a time,

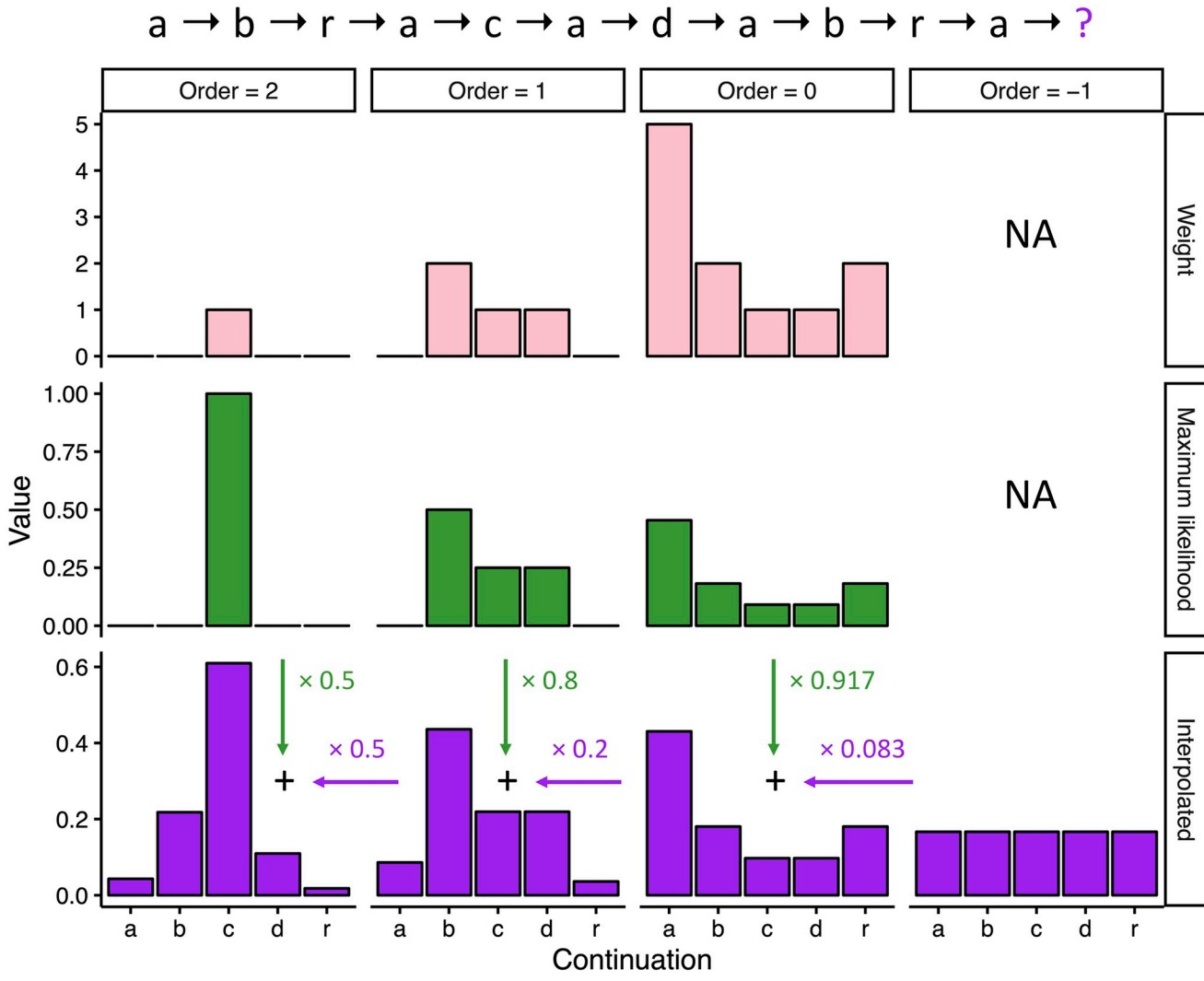

**Fig 11. Illustration of the interpolated smoothing mechanism.** This smoothing mechanism blends together maximum-likelihood $n$-gram models of different orders. Here the Markov order bound is two, the predictive context is "abracadabra", and the task is to predict the next symbol. Columns are identified by Markov order; rows are organized into weight distributions, maximum-likelihood distributions, and interpolated distributions. Maximum-likelihood distributions are created by normalizing the corresponding weight distributions. Interpolated distributions are created by recursively combining the current maximum-likelihood distribution with the next-lowest-order interpolated distribution. The labelled arrows give the weight of each distribution, as computed using escape method "A". The "Order = −1" column identifies the termination of the interpolated smoothing, and does not literally mean a Markov order of −1.

**Table 3. Summary of PPM-Decay hyperparameters.**

| Symbol | Name | Description |
|--------|------|-------------|
| $m_{max}$ | Markov order bound | Maximum length of conditioning context |
| $t_b$ | Temporal buffer capacity | Time after which observation is expunged from buffer |
| $n_b$ | Itemwise buffer capacity | Maximum number of symbols that can fit in buffer |
| $w_0$ | Buffer weight | Weight of $n$-gram while in buffer |
| $t_{0.5}$ | Half life | Half life of the exponential-decay phase |
| $w_1$ | Initial post-buffer weight | Weight of $n$-gram immediately after leaving buffer |
| $w_\infty$ | Asymptotic post-buffer weight | Weight of $n$-gram as time tends to infinity |
| $\sigma_\epsilon$ | Retrieval noise | Scale parameter for the retrieval noise distribution |

and then tries to predict the next symbol. The highest-order interpolated distribution, at a Markov order of two, is created by averaging the order-2 maximum-likelihood distribution with the order-1 interpolated distribution, which is itself created by averaging the order-1 maximum-likelihood distribution with the order-0 interpolated distribution. The resulting interpolated distribution combines information from maximum-likelihood models at every order.

We have implemented the resulting model in a freely available R package, "ppm", the core of which is written in C++ for speed. With this package, it is possible to define a PPM-Decay model customized by the eight hyperparameters summarized in Table 3. The package also supports simpler versions of PPM-Decay, where (for example) the buffer functionality is disabled but the exponential-decay functionality is preserved. The resulting models can then be evaluated on arbitrary symbolic sequences. The package may be accessed from its open-source repository at https://github.com/pmcharrison/ppm or its permananent archive at https://doi.org/10.5281/zenodo.2620414.

## Musical corpora

**Popular corpus.** This corpus was derived from the McGill Billboard corpus of [51], a dataset of popular music sampled from the Billboard 'Hot 100' charts between 1958 and 1991. The sampling algorithm was designed such that the composition dates should be approximately uniformly distributed between 1958 and 1991, and such that composition popularity should be approximately uniformly distributed across the range of possible chart positions (1–100). Having sampled 1,084 compositions with this algorithm, [51] had expert musicians transcribe the underlying chord sequences of these compositions. These transcriptions took a textual format, where each chord was represented as a combination of a root pitch class (e.g. 'Ab') and a chord quality (e.g. 'maj'). For example, the following text represents the beginning of 'Night Moves' by Bob Seger:

`| Ab:maj | Ab:maj . . Gb:maj | Db:maj | Db:maj . . Gb:maj |`

As is common in harmonic analyses, these transcriptions characterize chords in terms of their constituent *pitch classes*. A pitch class is an equivalence class of pitches under *octave transposition*; octave transposition means shifting a pitch by twelve semitones, which is equivalent to multiplying (or dividing) its fundamental frequency by a power of two.

This 'root + chord quality' representation is intuitive for performing musicians, but it is problematic for cognitive modeling in that the chord root is a subjective music-theoretic construct. We therefore translated these textual representations into sequences of *pitch-class chords*, defined as the combination of a bass pitch class with a set of non-bass pitch classes [114]. We performed this translation using the chord dictionary from the *hrep* software

package [114]. For this and the following corpora, an integer chord encoding was then derived by enumerating each unique chord observed in the respective corpus.

Harmonic analyses often do not systematically differentiate between one long chord and several repetitions of the same chord. In this and the following corpora we therefore collapsed consecutive repetitions of the same chord into single chords, as well as omitting all explicitly marked section repeats from the original transcriptions.

At the time of writing, only part of the Billboard corpus had been publicly released, the remainder being retained for algorithm evaluation purposes. Here we used the 739 transcriptions available at the time of writing, having removed transcriptions corresponding to duplicate compositions.

Fig 3A shows the resulting transcription for the first eight chords of 'Night Moves'. The full corpus is available in the *hcorp* R package alongside the other two musical corpora used in this paper (https://doi.org/10.5281/zenodo.2545754).

**Jazz corpus.** This corpus was derived from the iRb corpus of [52], a dataset of lead sheets for jazz compositions as compiled from an Internet forum for jazz musicians. Broze and Shanahan converted these lead sheets into a textual representation format termed `**jazz`, which (similar to the McGill Billboard corpus) expresses each chord as a combination of a root pitch class and a chord quality, alongside its metrical duration expressed as a number. For example, the following text represents the beginning of 'Thanks for the Memory' by Leo Robin:

```
2G:min7
2C7
=
1F6
=
2F6
2F#o7
=
4G:min7
4C7
2F6
=
2F#o7
2G:min7
=
2Ao7
2B-6
=
```

As with the popular music corpus, we translated these textual representations into sequences of pitch-class chords using the chord dictionary from the *hrep* package [114], and eliminated consecutive repetitions of the same chord. Fig 3B shows the result for the first eight chords of 'Thanks for the Memory'.

**Bach chorale corpus.** This corpus was derived from the '371 chorales' dataset from the KernScores repository [53]. This dataset comprises four-part chorale harmonizations by J. S. Bach, as collected by his son C. P. E. Bach and his student Kirnberger, and eventually digitally encoded by Craig Sapp. The 150th chorale harmonization is omitted from Sapp's dataset as it is not in four parts, leaving 370 chorales in total. This dataset uses the `**kern` representation scheme [115], designed to convey the core semantic information of traditional Western music notation. For example, the following text represents the first two bars of the chorale harmonization 'Mit Fried und Freud ich fahr dahin':

```
4D    4F    4A    4d
=1    =1    =1    =1
4C#   4A    4e    4a
4D    4d    4f    4a
4E    4B    4e    4g
8F#L        8AL   8dL   4dd
8G#J        8BJ   8eJ   .
=2    =2    =2    =2
4A    8cnXL       8eL   4ccnX
.     8dJ   8f#J        .
4E    8eL   4g#   4b
.     8dJ   .     .
4AA;        4c;   4e;   4a;
4E    [4c   4g    4cc
=3    =3    =3    =3
```

We derived chord sequences from these $^{**}$kern representations by applying the harmonic analysis algorithm of [54], which selects from a dictionary of candidate chords using a template-matching procedure. Here we used an extended version of this template dictionary, described in Table 4.

We computed one chord for each quarter-note beat, reflecting the standard harmonic rhythm of the Bach chorale style, and collapsed consecutive repetitions of the same chord into one chord, as before. Fig 3C shows the result for the first eight chords of the chorale harmonization 'Mit Fried und Freud ich fahr dahin'.

## Behavioral experiment

**Stimuli and procedure.** Each stimulus comprised a sequence of tones, with each tone gated on and off with 5-ms raised cosine ramps. Tone frequencies were drawn from a pool of 20 values equally spaced on a logarithmic scale between 222 Hz and 2,000 Hz. Tone length was always constant within a given trial and across trials in a block. Across blocks, three different

**Table 4. The dictionary of chord templates used in constructing the Bach chorale corpus.**

| Pitch classes | Label | Weight |
|---|---|---|
| [0, 4, 7, 11] | maj7 | 0.2 |
| [0, 3, 7, 10] | min7 | 0.2 |
| [0, 4, 8] | aug | 0.02 |
| [0, 7] | no3 | 0.05 |
| [0, 7, 10] | min7no3 | 0.05 |
| [0, 4, 7] | maj | 0.436 |
| [0, 7, 4, 10] | dom7 | 0.219 |
| [0, 3, 7] | min | 0.194 |
| [0, 3, 6, 9] | dim7 | 0.044 |
| [0, 3, 6, 10] | hdim7 | 0.037 |
| [0, 3, 6] | dim | 0.018 |

Each row identifies a different template. Each template comprises a set of pitch classes, expressed relative to the chord root. Applied to a collection of pitch classes within a harmonic segment, Pardo and Birmingham's [54] algorithm evaluates each candidate template with the respect to each of the 12 possible chord roots, and selects the template and root combination that best reflect the pitch-class content of the harmonic segment. Ties are broken using the 'weight' attribute; templates with higher weights are given priority.

tone durations were used (25, 50 and 75 ms). Individual stimuli ranged in length between 117 and 160 tones and in duration between 3,250 and 11,025 ms.

Four stimulus types were defined: 'CONT', 'STEP', 'RAND', and 'RAND-REG'. CONT and RAND trials contained no section change: CONT trials constituted one repeated tone of a given frequency, and RAND trials constituted randomly sampled tones from the full frequency pool, with the constraint that final tone counts were balanced by the end of the stimulus. STEP and RAND-REG trials each contained exactly one section change, occurring between 80 and 90 tones after sequence onset. Each section of a STEP trial comprised one repeated tone of a given frequency, with the section change constituting a change in frequency. RAND-REG trials comprised an initial random section, constructed under the same constraints as RAND trials, followed by a REG section constituting repeated iterations of a sequence of tones sampled randomly from the frequency pool without replacement. These repeating sequences comprised either 10 or 20 tones, depending on the block, with the REG section always comprising at least three repeating cycles. All stimuli were generated anew at each trial, and RAND and RAND-REG sequences occurred equiprobably.

The experimental session was delivered in 6 blocks, each containing 80 stimuli of a given tone length and alphabet size (35 RAND-REG, 35 RAND, 5 STEP, and 5 CONT), with the inter-stimulus interval jittered between 700 and 1100 ms, and with block duration ranging between 5.7 and 17.4 minutes. The order of blocks was randomized across participants. Before starting, participants were familiarized with the task with a short training session comprising six short blocks of 12 trials each, representing the same conditions as the main experiment. Stimuli were presented with the PsychToolBox in MATLAB (9.2.0, R2017a) in an acoustically shielded room and at a comfortable listening level selected by each listener.

Participants were encouraged to detect the transition as fast as possible. Correspondingly, feedback about response accuracy and speed was delivered at the end of each trial. This feedback consisted of a green circle if the response fell between the first and the second cycle of the regularity, or before 400 ms from the change of tone in the STEP condition; for slower RTs, an orange circle was displayed.

The RAND-REG trials were of primary interest for our analyses. We used the STEP trials to estimate baseline response times, computed separately for each participant within each block using correct responses only, and normalized the RAND-REG response times by subtracting these baseline response times. We excluded all RAND-REG trials where the participant responded incorrectly, and interpreted RAND and CONT trials as foils for the change-detection task.

**Participants.** We collected data from 25 paid participants (20 females; mean age 24.17, *SD* age = 3.17). Data from two participants were discarded due to overly slow reaction times on the STEP condition (mean reaction time more than three standard deviations from the mean).

**Preprocessing reaction time data.** We discarded 530 trials where participants responded incorrectly, and then normalized each participant's reaction times by subtracting the mean reaction time to all correctly answered STEP trials in the same block. We then retained all RAND-REG trials where the normalized reaction times fell within two standard deviations from the mean for a given combination of participant, tone duration, and cycle length. This left 4,439 trials.

## Modeling reaction time data

We modeled participants' reaction times using the new PPM-Decay model presented in *Model*. We modeled each trial separately, resetting the model's memory after each trial.

We modeled participants' change detection processes using a non-parametric change-detection algorithm that sequentially applies the Mann-Whitney test to identify changes in a time series' location while controlling the false positive rate [69, 116]. We used the algorithm as implemented in the 'cpm' R package [116], setting the desired false positive rate to one in 10,000, and the algorithm's warm-up period to 20 tones.

For comparison with the participant data, we computed representative model reaction times for each condition by taking the mean reaction time over all trials where the model successfully detected a transition, excluding any trials where the model reported a transition before the effective transition (this resulted in excluding 0.32% of trials). We used R and C++ for our data analyses [117]; our PPM-Decay implementation is available at https://github.com/pmcharrison/ppm and https://doi.org/10.5281/zenodo.2620414. Raw data, analysis code, and generated outputs are archived at https://doi.org/10.5281/zenodo.3603058.

## Acknowledgments

The authors are grateful to Bastiaan van der Weij for providing useful feedback on the manuscript, and to Ji-Myoung Choi for providing useful feedback on the *ppm* package.

## Author Contributions

**Conceptualization:** Peter M. C. Harrison, Roberta Bianco, Maria Chait, Marcus T. Pearce.

**Data curation:** Peter M. C. Harrison, Roberta Bianco.

**Formal analysis:** Peter M. C. Harrison, Roberta Bianco.

**Funding acquisition:** Maria Chait, Marcus T. Pearce.

**Investigation:** Peter M. C. Harrison, Roberta Bianco.

**Methodology:** Peter M. C. Harrison, Roberta Bianco, Maria Chait, Marcus T. Pearce.

**Project administration:** Maria Chait, Marcus T. Pearce.

**Resources:** Peter M. C. Harrison, Roberta Bianco, Maria Chait, Marcus T. Pearce.

**Software:** Peter M. C. Harrison.

**Supervision:** Maria Chait, Marcus T. Pearce.

**Validation:** Peter M. C. Harrison, Roberta Bianco, Maria Chait, Marcus T. Pearce.

**Visualization:** Peter M. C. Harrison.

**Writing – original draft:** Peter M. C. Harrison, Roberta Bianco.

**Writing – review & editing:** Peter M. C. Harrison, Roberta Bianco, Maria Chait, Marcus T. Pearce.

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
