## [Decision Letter · Decision Letter 0]

12 May 2020

Dear Mr Harrison,

Thank you very much for submitting your manuscript "PPM-Decay: A computational model of auditory prediction with memory decay" for consideration at PLOS Computational Biology.

As with all papers reviewed by the journal, your manuscript was reviewed by members of the editorial board and by several independent reviewers. In light of the reviews (below this email), we would like to invite the resubmission of a significantly-revised version that takes into account the reviewers' comments.

Dear authors,

Both reviewers agreed that your work has merit but 1) many points need clarification and 2) it might be less novel than advertised. You will want to try the models suggested by reviewer 1 and broaden your discussion to provide fair comparisons with the computational methods used in linguistics.

Best wishes,

Frederic.

We cannot make any decision about publication until we have seen the revised manuscript and your response to the reviewers' comments. Your revised manuscript is also likely to be sent to reviewers for further evaluation.

Sincerely,

Frédéric E. Theunissen

Associate Editor

PLOS Computational Biology

Daniele Marinazzo

Deputy Editor

PLOS Computational Biology

Dear authors,

Both reviewers agreed that your work has merit but 1) many points need clarification and 2) it might be less novel than advertised. You will want to try the models suggested by reviewer 1 and broaden your discussion to provide fair comparisons with the computational methods used in linguistics.

Best wishes,

Frederic.

Reviewer's Responses to Questions

**Comments to the Authors:**

Reviewer #1: The authors present essentially an n-gram model that more heavily weights recent data and show that it out-performs an unweighted model on a simple toy dataset, and then that it does a better job of predicting note sequences in a music dataset. Then, the authors model experimental data in which participants listen to tone sequence and must detect a shift to randomness. They compare a variety of different versions of this model, including PPM, PPM+decay, and PPM+decay+retrieval noise with varying item buffers, and find that models with retrieval noise are a substantially better fit to human performance.

The results and model seem reasonable, but there is a huge literature on related methods, mostly in computational linguistics, that seem worth connecting to in some way. For instance, there are many Hidden Markov Model inference schemes that take observations of discrete sequences and try to construct a markov model to make predictions, and in this inference there are often naturally arising forms of memory decay when making predictions. There is also related work on clustering input sequence (for a historical reference, see Carter "Improving Language Models by Clustering Training Sentences").

Finally, there are versions of this kind of decayed n-gram model sometimes under the name of "language model adaptation" (I don't know all of them, but a start in the literature are "Language model adaptation using mixtures and an exponentially decaying cache", "Time-Sensitive Language Modelling for Online Term Recurrence Prediction", and "Statistical language model adaptation: review and perspectives")

I think that before I could recommend acceptance, the work would have to be brought in contact with this broader literature. I don't think the existence of these kinds of models really impedes the authors' ability to publish, I just would not want them to miss the opportunity to connect to a broader modeling literature.

Additional comments:

- It strikes me that PPM-decay shares a lot of similarities with Lempel-Ziv compression (or any variants, like LZW) in that repeated data is given a shorter code word. I think it would strengthen the paper substantially to test out some of these other compression schemes to be sure that PPM-decay is not just doing some kind of approximation scheme to another compression scheme.

- Exponential weighting is interesting, but language (and probably music) follow a power-law time series dependency. The authors might be interested in Anderson & Schooler "Reflections of the environment in memory" who look at the gap between word repetitions and show they are power-law distributed. It seems like a similar empirical approach to quantify the distribution of repetitions of subsequnces could be useful here, and I'd be surprised if it wasn't power law.

- Why is the PPM-Decay model evaluated based on prediction accuracy instead of a more standard NLP measure like perplexity (2^surprisal)? I like having accuracy since it's intuitive, but I think perplexity is also important to include since it penalizes mistakes differently.

- The paper needs more details on how the PPM-decay model works -- I only found these in the supplemental. For instance, it would be useful to put some description of the smoothing into the main text.

- Why doesn't the model use state-of-the-art smoothing techniques? See, e.g., Chen & Goodman "An empirical study of smoothing techniques for language modeling" is a classic and very good reference on this.

- It seems like it would be interesting to compare a buffer based on bits of information rather than items -- using bits is popular both in visual memory studies and in psycholinguistics, and bit-vs-item based accounts make different predictions. Because the authors have a probabilistic model, it seems like they are in a good position to do this.

- It's worth thinking about some places where this model may not make the right predictions -- for instance, if you have a memorized piece of music or you tend to remember a particular style of music for long periods of time. Is PPM-decay just an approximation to some fancier form of clustering genres?

- What license is the software provided under?

Reviewer #2: The manuscript presents an extension to the Prediction by Partial Matching (PPM) model, to formally include the notion of memory limitation. Three distinct tasks are described: the recovery of a time-varying statistical structure in artificially-generated sequences of symbols; the effect of exposure to predict the structure of musical scores; the prediction of behavioral performance in a regularity-detection task for a novel set of experimental data. In all cases, the core PPM idea successfully adapted to the task by adding some form of memory limitation. Part 1 shows that imposing an exponential decay on past information is beneficial to adapt to changing statistics. Part 2 introduces a non-zero asymptote to the decay and shows that this asymptote (representing long-term learning) has a different weight for predicting different musical genres. For part 3, a combination of a short, perfect memory store followed by exponential decay and noise is introduced, resembling some models of perceptual memory.

The manuscript provides a formal test of a natural extension to the PPM model. By adding memory limitations, the model is no more limited to evaluating the limit-case of an observer with perfect memory of the past, but has much greater flexibility to be fit to different tasks and situations. This is an important advance, which should be of clear interest to the readership of the journal. Even though subparts of the manuscript may at times feel like a paper within a paper, there is still a logic in having all three studies in the same place. The fact that the precise form of memory limitations needed to fit the tasks turned out to be different in all three cases initially bothered me, but in part thanks to the Discussion, I am more convinced that this could be seen as a strength of the approach, as the model can be used as a tool to highlight specific aspects of memory involved in specific tasks.

I have very little comments with parts 1 and 2, which I found highly convincing. I thought that the artificial example of part 1 made the point clearly, and while results were perhaps unsurprising, the bigger idea that forgetting helps in changing environments is interesting. The conclusion reached in part 2, on differences of long-term vs short-term regularity with genre, would be of great interest to musicologists, and could have been the bulk of an independent manuscript in a different journal. I have a few more comments for part 3, with some issues that may need addressing prior to publication.

Specific comments

L362. The motivation given to gather new experimental data (and not simply model Barascud et al. (2016)) is to test whether auditory memory is limited in terms of duration or capacity, citing ‘historic’ work supporting the capacity idea. I think this very interesting question is not limited to historic work, and furthermore that there may be yet other alternatives to describe the limits of auditory memory. For instance, the work of Demany and colleagues suggests almost perfect capacity and essentially no decay seconds in some tasks (Demany et al., 2008, Psych Science, see also the book chapter Demany and Semal 2007). Kang et al., 2017, J Acoust Soc Am, also outline alternative ideas based on randomly sampled pattern motifs. As the precise form of memory limitation is central to the argument made by the new task and subsequent modeling, maybe a more complete survey of the available alternatives would be useful?

L379. Remarkably for human behavior, but unfortunately for the manuscript’s purpose, performance in the new task was very near ceiling for all conditions and participants. Doesn’t this mean that there must exist a memory store of some kind that contained enough information for the task to be performed perfectly? Operationaly, would there be a statistics extracted from the PPM-decay model put forward that predicts perfect performance in all cases?

L382. As is commonly done in behavioral research, reaction times are considered as a surrogate measure of ‘performance above ceiling’. So, RTs are chosen as the target for the modeling. However, I think this general idea need a very careful justification here, as there may be other models/strategies that lead to different RTs without requiring any form of memory decay, thus defeating the main point of the modeling. For instance, what if it simply took longer to scan a perfect but longer memory trace?

I think what is missing for me is an explicit description of what it would take for an observer using the PPM-decay model to achieve both perfect performance for all conditions and slower RTs for some conditions. Such an explicit statement of the underlying assumptions could also help discussing possible alternative interpretations.

L398. The analysis uses differences of differences to make a discussion point. Why not use an ANOVA on the whole results and look for interactions?

Fig 6. The proposed form of the decay function appended to the PPM is quite complex and implies four a priori choices: a perfect buffer of constant capacity that is fine-tuned to each stimulus duration; an abrupt drop at the end of this buffer; an exponential decay; additive noise. On the plus side, the logic for each refinement is well discussed in the text. However, one has to wonder how much of these choices were really constrained by the data. Would there be other, very different strategies that would fit the data? Or different parameter sets with the same model structure? This is I think important to ask, because if the modeling is to provide evidence for the nature of the underlying cognitive processes, one would want to know how unique and specific is a parameter set that happens to fit the task. The text mentions that the model was manually tuned, maybe some sort of parameter grid search would be useful?

L 428: setting a buffer length of 15 items seems ad-hoc, given that the experiment tested sequences of 10 and 20 items and observed a drop in performance between the two. How confident are we that the value is not tied to the experimental details?

L439. The noise added to the model representation is described as having a sigma of 0.8. Is the unit the same as the weights, and if so, how to interpret such a value given that the maximum possible weight value is 1?

Methods: There were in fact more conditions run than described in main text, which is not great given that some conditions (CONT, STEP) were used as normalization for all the data presented. Without breaking the flow of the main story, I think it would be useful for the reader to form a clear picture of the task before delving into the methods.

**Have all data underlying the figures and results presented in the manuscript been provided?**

Reviewer #1: Yes

Reviewer #2: Yes

PLOS authors have the option to publish the peer review history of their article (what does this mean?). If published, this will include your full peer review and any attached files.

Reviewer #1: No

Reviewer #2: No
---

## [Decision Letter · Decision Letter 1]

4 Sep 2020

Dear Mr Harrison,

We are pleased to inform you that your manuscript 'PPM-Decay: A computational model of auditory prediction with memory decay' has been provisionally accepted for publication in PLOS Computational Biology.

Best regards,

Frédéric E. Theunissen

Associate Editor

PLOS Computational Biology

Daniele Marinazzo

Deputy Editor

PLOS Computational Biology

Reviewer's Responses to Questions

**Comments to the Authors:**

Reviewer #1: The authors have undertaken a revision that addresses the concerns I raised in my original review. In particular, they now do a much better job of connecting the model to prior models and this helps to situate it within the literature. The figures are clear and nice. I appreciated their detailed responses in the cover letter. I'm happy to recommend publication.

(One comment: In the analysis for Experiment 3, I wonder if the authors could say a bit more about the buffer lengths that should be expected given prior literature. The buffer size seems to mainly be treated as a free parameter, but it seems like it could be informed by studies of auditory memory? What exactly does e.g. 10 or 15 items correspond to in, for instance, speech processing (is an item a phoneme? a morpheme? a word?))

Reviewer #2: The additions to the manuscript have clarified and addressed all of the issues that were raised in my previous review. I understand that Reviewer 1 raised very different concerns, but at least from my perspective, I do not have any further comments. Congratulations to the authors for what could be a very useful model in a range of different settings.

**Have all data underlying the figures and results presented in the manuscript been provided?**

Reviewer #1: Yes

Reviewer #2: Yes

PLOS authors have the option to publish the peer review history of their article (what does this mean?). If published, this will include your full peer review and any attached files.

Reviewer #1: No

Reviewer #2: No

---

## [Editor Report · Acceptance letter]

28 Oct 2020

PCOMPBIOL-D-20-00359R1 

PPM-Decay: A computational model of auditory prediction with memory decay

Dear Dr Harrison,

I am pleased to inform you that your manuscript has been formally accepted for publication in PLOS Computational Biology. Your manuscript is now with our production department and you will be notified of the publication date in due course.

With kind regards,

Matt Lyles
